# Multiancestry exome sequencing reveals *INHBE* mutations associated with favorable fat distribution and protection from diabetes

Parsa Akbari[1,13], Olukayode A. Sosina[1,13], Jonas Bovijn[1,13], Karl Landheer[2], Jonas B. Nielsen[1], Minhee Kim[1], Senem Aykul[1], Tanima De[1], Mary E. Haas[1], George Hindy[1], Nan Lin[1], Ian R. Dinsmore[3], Jonathan Z. Luo[3], Stefanie Hectors[2], Benjamin Geraghty[1], Mary Germino[2], Lampros Panagis[2], Prodromos Parasoglou[2], Johnathon R. Walls[2], Gabor Halasz[2], Gurinder S. Atwal[2], Regeneron Genetics Center*, DiscovEHR Collaboration*, Marcus Jones[1], Michelle G. LeBlanc[1], Christopher D. Still[4], David J. Carey[4], Alice Giontella[5,6], Marju Orho-Melander[5], Jaime Berumen[7], Pablo Kuri-Morales[7,8], Jesus Alegre-Díaz[7], Jason M. Torres[9,10], Jonathan R. Emberson[9,10], Rory Collins[10], Daniel J. Rader[11], Brian Zambrowicz[2], Andrew J. Murphy[2], Suganthi Balasubramanian[1], John D. Overton[1], Jeffrey G. Reid[1], Alan R. Shuldiner[1], Michael Cantor[1], Goncalo R. Abecasis[1], Manuel A. R. Ferreira[1], Mark W. Sleeman[2], Viktoria Gusarova[2], Judith Altarejos[2], Charles Harris[2], Aris N. Economides[1,2], Vincent Idone[2], Katia Karalis[1], Giusy Della Gatta[1], Tooraj Mirshahi[4], George D. Yancopoulos[2], Olle Melander[5,12], Jonathan Marchini[1], Roberto Tapia-Conyer[8,13], Adam E. Locke[1,13], Aris Baras[1,13] ✉, Niek Verweij[1,13] & Luca A. Lotta[1,13] ✉

Body fat distribution is a major, heritable risk factor for cardiometabolic disease, independent of overall adiposity. Using exome-sequencing in 618,375 individuals (including 160,058 non-Europeans) from the UK, Sweden and Mexico, we identify 16 genes associated with fat distribution at exome-wide significance. We show 6-fold larger effect for fat-distribution associated rare coding variants compared with fine-mapped common alleles, enrichment for genes expressed in adipose tissue and causal genes for partial lipodystrophies, and evidence of sex-dimorphism. We describe an association with favorable fat distribution ($p = 1.8 \times 10^{-09}$), favorable metabolic profile and protection from type 2 diabetes (~28% lower odds; $p = 0.004$) for heterozygous protein-truncating mutations in *INHBE*, which encodes a circulating growth factor of the activin family, highly and specifically expressed in hepatocytes. Our results suggest that inhibin βE is a liver-expressed negative regulator of adipose storage whose blockade may be beneficial in fat distribution-associated metabolic disease.

A full list of affiliations appears at the end of the paper. *Lists of authors and their affiliations appear at the end of the paper.
✉ e-mail: aris.baras@regeneron.com; luca.lotta@regeneron.com

The ability to store excess calories in adipose tissue in the form of triglycerides is essential to metabolic health in humans[1–6] and the distribution of fat in the body is a major risk factor for cardiometabolic disease[6–9], independent of overall adiposity.

In individuals from different world regions, a higher waist-to-hip circumference ratio (WHR), a simple proxy-measure of the relative abundance of abdominal to gluteofemoral fat, is strongly associated with higher incidence of cardiovascular disease and diabetes[7,8,10], independent of body mass index (BMI).

While fat distribution is a major epidemiological risk factor accounting for a large share of the global morbidity of cardiometabolic disease, there is a lack of therapeutic options to modify improper fat storage. A deeper understanding of the genetic basis of fat distribution and its relationships with disease may translate into new therapeutic approaches.

In Mendelian genetic studies, rare variants in *PPARG*[11], a transcription factor and master-regulator of adipocyte differentiation, and in six other genes have been associated with familial partial lipodystrophy (FPLD)[12,13]. Partial lipodystrophies are extreme forms of centripetal body fat distribution characterized by the inability to expand peripheral adipose storage, with deposition of excess calories as ectopic fat in the liver, leading to insulin resistance, diabetes and vascular disease[12,13]. It has been suggested that similar mechanisms are at play in more subtle forms of cardiometabolic disease of unknown genetic etiology in the general population[1].

Consistent with this hypothesis, genome-wide association studies (GWAS) have successfully identified hundreds of common genetic variants associated with fat distribution and provided evidence of strong etiologic relationships with diabetes and coronary disease[14–19].

However, the demonstration of key underlying mechanisms for fat distribution-associated disease and their molecular determinants have been elusive, contributing to challenges in identifying therapeutically modifiable pathways. In particular, the excessive deposition of hepatic fat has been proposed to play a central role in the link between body shape and disease[1]. Hepatic steatosis is a driver of insulin resistance, dyslipidemia and nonalcoholic steatohepatitis (NASH), a highly prevalent and fast-growing cause of global morbidity and mortality[20]. Genetic variants associated with accumulation of fat in the abdominal cavity or with lower levels of fat deposition in gluteofemoral regions have been hypothesized to cause hepatic steatosis as key mechanistic steps towards type 2 diabetes and coronary disease[16,21]. However, it has not been possible to demonstrate these genetic mechanisms and pinpoint their molecular effectors due to a lack of large genomic databases linked to refined measures of liver fat, inflammation and fibrosis.

Here, we tackled these outstanding questions with human genetic studies centered around the exome sequencing of 618,375 individuals across five ancestries. This approach may identify naturally occurring loss-of-function (LOF) alleles that protect from disease[22,23], a type of genetic association which has informed therapeutic target identification in a growing number of examples[24,25]. We also combined exome sequencing with common-variant polygenic scores and with refined measures of liver fat and inflammation, to study the role of liver health in fat distribution-associated cardiometabolic disease (Fig. 1).

## Results
### Exome-wide associations with body fat distribution
We leveraged multi-ancestry exome-sequencing of 618,375 individuals from three population-based cohorts in the UK, Sweden and Mexico ("Methods", Supplementary Data 1), including 160,058 non-European individuals. We estimated associations with fat distribution, measured as BMI-adjusted WHR, for the burden of rare nonsynonymous variants in each gene in the genome, conditional upon 868 common variants (listed in Supplementary Data 2) identified by fine-mapping of GWAS signals in the same participants ("Methods").

Sixteen genes were associated with fat distribution at exome-wide statistical significance (inverse-variance weighted [IVW] meta-analysis $p < 3.6 \times 10^{-7}$; Table 1, Supplementary Fig. 1), with consistent effect estimates across ancestries (heterogeneity $I^2$ below 75%[26] for each association; Supplementary Data 3). Rare predicted-deleterious coding alleles in *PLXND1* and *CD36* were 2.5- and 4.5-fold enriched in American ancestry individuals relative to Europeans, providing critical evidence implicating these genes (Supplementary Data 3). A median of 296 (interquartile range, 187–428; Table 1 and Supplementary Data 4) distinct rare coding variants per gene contributed to the gene-burden exposures. Effect estimates were on average six-fold larger for gene-burden associations than for the 868 fine-mapped common-variant signals identified in the same individuals (Fig. 2).

Gene-burden associations had near perfect correlation in a BMI-unadjusted analysis (Pearson correlation, 0.99; $p = 9.9 \times 10^{-13}$; Supplementary Fig. 2), indicating that collider bias[27,28] due to BMI adjustment did not drive the identification of these genes. To assess the potential influence of skeletal phenotypes on these associations, we performed sensitivity analyses adjusted for height or estimated bone mineral density, which yielded near-identical associations as the main analysis (height adjustment Pearson correlation, 1; $p = 6.2 \times 10^{-31}$; estimated bone mineral density adjustment Pearson correlation, 1; $p = 4.6 \times 10^{-18}$; Supplementary Fig. 2). We also showed near-identical estimates with a nonlinear adjustment for body fat mass measured by electrical bioimpedance (Pearson correlation, 1; $p = 1.3 \times 10^{-16}$; Supplementary Fig. 2), ruling out an influence of nonlinear relationships with overall body adiposity on the associations.

To further corroborate that the identified associations reflect a difference in fat distribution, we studied visceral-to-gluteofemoral fat ratio derived from whole-body magnetic resonance imaging (MRI), a "gold-standard" measure available in a subset of 38,880 people (i.e. ~6% of the discovery sample; Supplementary Data 5, Supplementary Fig. 3). Association estimates showed 94% directional concordance between BMI-adjusted WHR and visceral-to-gluteofemoral fat ratio (expected proportion under null assumption, 50%; two-way binomial for observed proportion $p = 5.2 \times 10^{-4}$) and gene-burden associations were highly consistent between the two traits (beta in SD units of visceral-to-gluteofemoral fat ratio per 1 SD higher BMI-adjusted WHR via the 16 genes, 1.30; 95% confidence interval [CI], 1.04, 1.56; $p = 9.3 \times 10^{-23}$; Supplementary Fig. 3). We observed similar consistency for a polygenic score based on 202 WHR-associated common variants[16] (beta in SD units of visceral-to-gluteofemoral fat ratio per 1 SD higher BMI-adjusted WHR via the polygenic score, 1.09; 95% CI, 1.04, 1.14; $p = 2.2 \times 10^{-360}$).

For four of 16 genes (*ACVR1C*, *CALCRL*, *PLIN1*, *PDE3B*), rare coding variant associations with BMI-adjusted WHR have been previously reported at the genome- and exome-wide significance thresholds used here[17,18,29] (Supplementary Data 6), while *PDE3B* rare coding alleles have been associated with BMI[22]; the remaining 12 associations had not been reported in previous studies.

Two of the 16 genes (*PPARG*, *PLIN1*) were causative genes for familial partial lipodystrophies (FPLDs), which are Mendelian forms of extreme fat distribution (Supplementary Data 7; fold-enrichment, 554; 95% confidence interval [CI], 49 to 3623; Fisher's exact test $p = 1.3 \times 10^{-5}$). The burden of rare pLOF variants or rare pLOF plus predicted deleterious missense variants in six of seven known FPLD genes (all except *AKT2*) showed a nominal association with fat distribution ($p < 0.05$; Supplementary Data 7). Interestingly, in our analysis, *PLIN1* pLOF variants were associated with lower BMI-adjusted WHR and larger hip circumference, a phenotype that is opposite of that observed[30] in individuals with C-terminal frameshift variants in *PLIN1* and autosomal dominant FPLD type 4 (Supplementary Result 1). This suggests that the lipodystrophy phenotype observed in FPLD type 4 might be due to a peculiar alteration in PLIN1 function caused by those specific C-terminal frameshift variants and

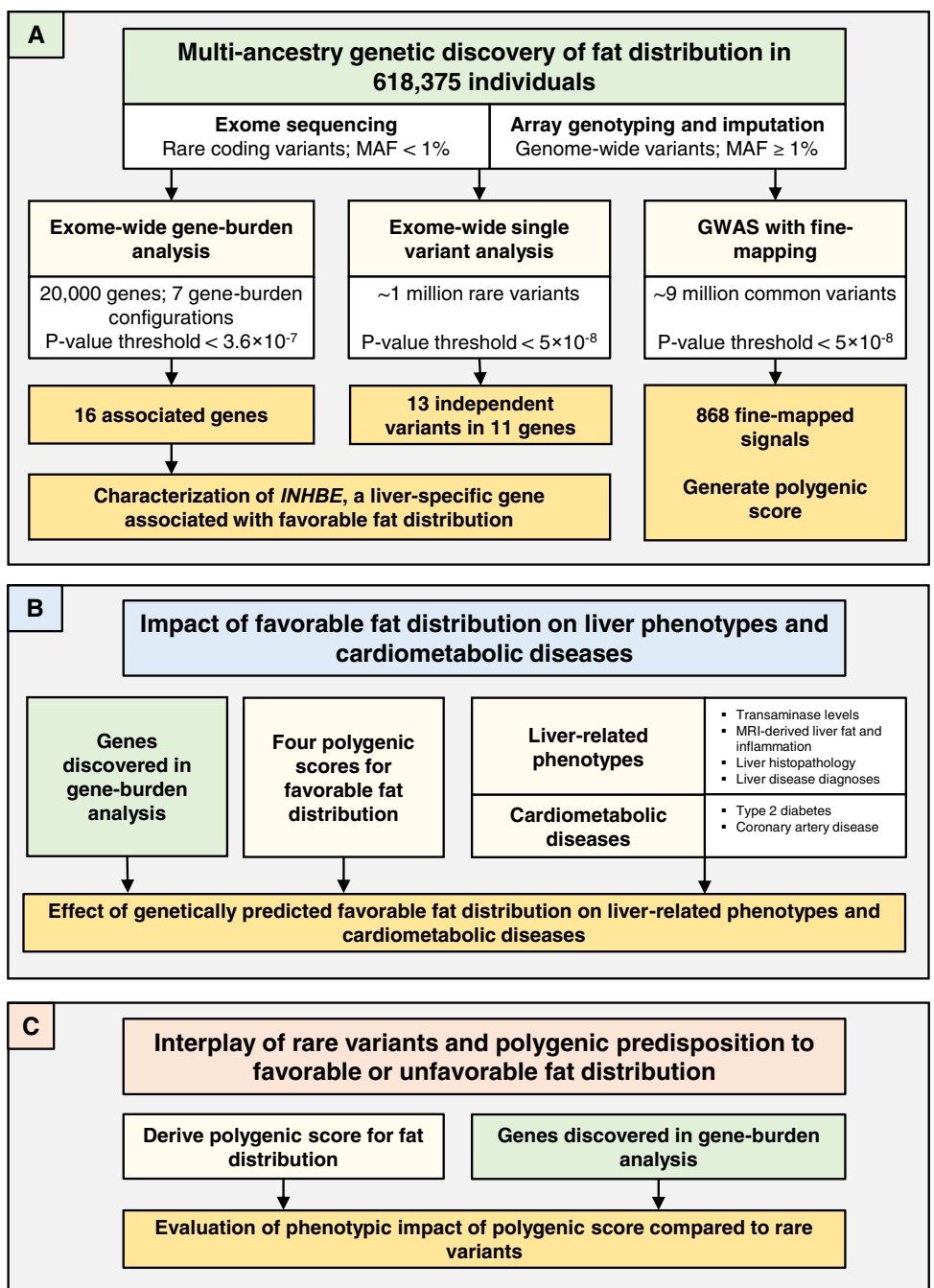

**Fig. 1 | Study overview.** We performed three major groups of analyses, illustrated in (**A**–**C**). **A** outlines the multi-ancestry genetic discovery analysis for BMI-adjusted WHR. **B** summarizes association analyses for genetically determined fat distribution, liver health parameters and cardiometabolic disease. **C** outlines the evaluation of common and rare variant interplays in fat distribution. MAF minor allele frequency, GWAS genome-wide association study, MRI magnetic resonance imaging.

not a simple heterozygous loss of *PLIN1* function (i.e. haploinsufficiency).

We observed enrichment for genes highly expressed in subcutaneous adipose tissue, the specialized energy storage tissue of the body (Supplementary Fig. 4; Enrichment Wald test $p = 7.3 \times 10^{-4}$), and, to a lesser degree, visceral adipose tissue (Supplementary Fig. 4; Enrichment Wald test $p = 1.1 \times 10^{-3}$). For five of 16 genes, subcutaneous adipose was the highest expressing tissue across 48 tissue types, while one gene had highest expression in visceral adipose tissue (Supplementary Fig. 5). We estimated associations with hip and waist circumference, as proxy measures of gluteofemoral and abdominal fat respectively. Thirteen of 16 genes showed nominal associations with

hip circumference (IVW meta-analysis p < 0.05), while five genes were associated with waist (IVW meta-analysis $p < 0.05$; Supplementary Data 8).

In line with previous literature on fat distribution[14,18,19,31], we observed evidence of sex-interaction for eight of 16 genes ($p_{interaction} < 3.1 \times 10^{-3}$, Bonferroni correction for 16 genes at α = 0.05; Supplementary Data 9), with stronger associations in women for all eight genes. In sex-stratified discovery analyses for BMI-adjusted WHR, we identified two additional genes in the women-only analysis which showed no associations in men (*FGF1* and *MSR1;* Table 1; Supplementary Data 9).

To complement gene-burden analyses, we performed a rare single variant discovery analysis identifying 13 independently associated

**Table 1 | Associations with fat distribution in the exome-wide gene-burden analysis**

| Gene | Variants contributing to burden test | Genetic exposure, variant type; frequency cutoff in % | Beta (95% CI) per allele in SD units of BMI-adjusted WHR | P | AAF, fraction of 1 | Genotype counts (RR | RA | AA genotypes) |
|---|---|---|---|---|---|---|
| ACVR1C 2: 157526766 | 173 | pLOF plus deleterious missense (5/5); AAF <1% | −0.16 (−0.19, −0.12) | $3.1 \times 10^{-20}$ | 0.0025 | 615,316 | 3057 | 2 |
| CALCRL 2: 187343128 | 311 | pLOF plus deleterious missense (1/5); AAF <1% | −0.087 (−0.11, −0.06) | $1.5 \times 10^{-10}$ | 0.0038 | 613,641 | 4730 | 4 |
| PPARG 3: 12287367 | 327 | pLOF plus deleterious missense (1/5); AAF <1% | 0.14 (0.089, 0.18) | $1.3 \times 10^{-08}$ | 0.0012 | 616,856 | 1519 | 0 |
| STAB1 3: 52495337 | 970 | pLOF plus deleterious missense (5/5); AAF <0.1% | −0.065 (−0.086, −0.045) | $2.8 \times 10^{-10}$ | 0.0067 | 610,105 | 8262 | 8 |
| PLXND1 3: 129555174 | 1425 | pLOF plus deleterious missense (1/5); AAF <1% | −0.03 (−0.042, −0.019) | $7.3 \times 10^{-08}$ | 0.0231 | 589,953 | 28,329 | 93 |
| CD36 7: 80369574 | 525 | pLOF plus deleterious missense (5/5); AAF <1% | 0.048 (0.031, 0.066) | $6.8 \times 10^{-08}$ | 0.0090 | 607,219 | 11,126 | 30 |
| ABCA1 9: 104781001 | 880 | pLOF plus deleterious missense (5/5); AAF <1% | −0.056 (−0.074, −0.038) | $5.1 \times 10^{-10}$ | 0.0087 | 607,690 | 10,668 | 17 |
| AIFM2 10: 70098222 | 321 | pLOF plus deleterious missense (1/5); AAF <1% | 0.049 (0.036, 0.063) | $2.1 \times 10^{-12}$ | 0.0145 | 600,542 | 17,782 | 51 |
| PDE3B 11: 14643722 | 281 | pLOF plus deleterious missense (5/5); AAF <0.1% | −0.18 (−0.22, −0.15) | $1.4 \times 10^{-22}$ | 0.0020 | 613,713 | 2459 | 0 |
| INHBE 12: 57455320 | 29 | pLOF; AAF <1% | −0.17 (−0.22, −0.11) | $1.8 \times 10^{-09}$ | 0.0009 | 614,471 | 1096 | 1 |
| PLIN1 15: 89664364 | 118 | pLOF plus deleterious missense (5/5); AAF <1% | −0.2 (−0.23, −0.17) | $4.6 \times 10^{-32}$ | 0.0025 | 615,348 | 3021 | 6 |
| ANKRD12 18: 9136776 | 156 | pLOF; AAF <1% | 0.31 (0.22, 0.4) | $1.6 \times 10^{-11}$ | 0.0003 | 615,156 | 412 | 0 |
| PLIN4 19: 4502179 | 195 | pLOF; AAF <1% | 0.11 (0.079, 0.14) | $3.7 \times 10^{-13}$ | 0.0031 | 614,492 | 3874 | 9 |
| INSR 19: 7112254 | 215 | pLOF plus deleterious missense (5/5); AAF <1% | −0.075 (−0.094, −0.055) | $1.2 \times 10^{-13}$ | 0.0069 | 609,823 | 8514 | 38 |
| KEAP1 19: 10486119 | 396 | pLOF plus deleterious missense (1/5); AAF <1% | 0.089 (0.066, 0.11) | $3.4 \times 10^{-14}$ | 0.0051 | 612,099 | 6266 | 10 |
| SLC5A3 21: 34073569 | 191 | pLOF plus deleterious missense (5/5); AAF <1% | 0.06 (0.041, 0.078) | $4.7 \times 10^{-10}$ | 0.0077 | 608,903 | 9442 | 30 |
| **Women-only analysis** | | | | | | |
| FGF1 5: 142592177 | 153 | pLOF plus deleterious missense (1/5); AAF <1% | −0.083 (−0.11, −0.051) | $2.8 \times 10^{-07}$ | 0.0045 | 352,178 | 3183 | 6 |
| MSR1 8: 16107877 | 165 | pLOF plus deleterious missense (5/5); AAF <1% | −0.071 (−0.096, −0.045) | $9.8 \times 10^{-08}$ | 0.0068 | 350,564 | 4786 | 17 |

The table reports genes for which the burden of rare nonsynonymous variants was associated with BMI-adjusted WHR at exome-wide statistical significance (IVW meta-analysis, $p < 3.6 \times 10^{-7}$). Analyses were performed in 618,375 individuals from UKB, MDCS and MCPS. Effect sizes in ratio units can be obtained by multiplying the effect sizes in SD units by 0.08 ratio units. Genomic coordinates reflect chromosome and position in base pairs according to the Genome Reference Consortium Human Build 38. AAF was derived by dividing the number of alternative alleles observed for a particular gene-burden by the total number of all alleles observed for that gene-burden. P-values are from two-sided Z-tests from fixed-effect meta-analysis. CI confidence intervals, SD standard deviations, BMI-adjusted WHR waist-hip ratio adjusted for body mass index, P P-value, AAF alternative allele frequency, RR reference-reference homozygous genotype, RA reference-alternative heterozygous genotype, AA alternative-alternative homozygous genotype, pLOF predicted loss of function, Missense (5/5) missense variants predicted to be deleterious by 5 out of 5 in silico prediction algorithms, Missense (1/5) missense variants predicted to be deleterious by at least 1 out of 5 in silico prediction algorithms.

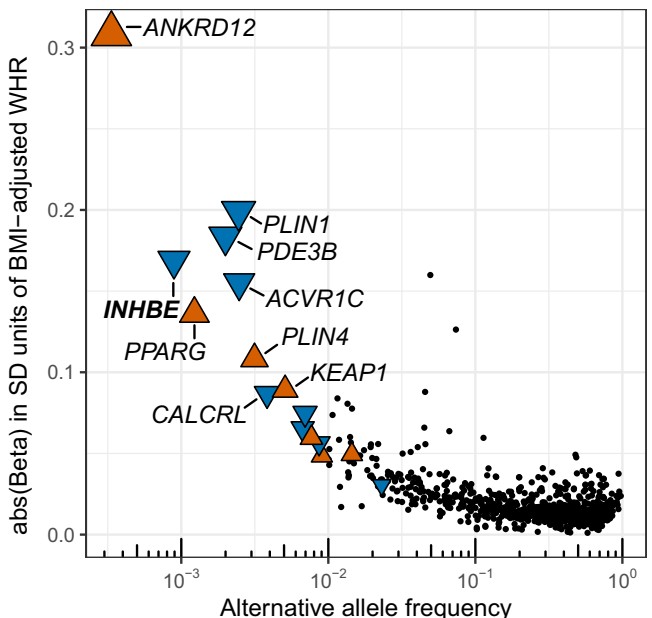

**Fig. 2 | Associations with BMI-adjusted WHR for common and rare alleles in the multi-ancestry analysis.** The 16 genes with exome-wide significant gene-burden associations are shown as colored triangles, with the triangles pointing upwards (orange) or downwards (blue) indicating associations with higher and lower BMI-adjusted WHR, respectively. The 868 fine-mapped common variants are indicated as black dots. The alternative allele frequency for each variant or gene-burden genotype is indicated on the x-axis. SD standard deviation, WHR waist to hip ratio, BMI body mass index.

variants in 11 genes (IVW meta-analysis $p < 5 \times 10^{-8}$; Supplementary Fig. 1 and Supplementary Data 10). These included a variant in a gene not highlighted in the gene-burden analysis nor in previous genetic studies: a Val136Ile missense variant in *GH1* (Supplementary Data 10).

In summary, we identified several associations with fat distribution for rare coding variants that (a) are robust in a variety of sensitivity analyses; (b) are highly correlated with a "gold-standard" fat distribution measure; (c) have large-effect sizes; (d) are enriched for genes highly expressed in adipose tissue and for causal genes of Mendelian forms of extreme fat distribution and (e) often exhibit sex-dimorphism.

**Loss of function in liver-specific *INHBE* is associated with favorable fat distribution and protection from metabolic disease**
We explored in depth the association with favorable fat distribution for rare pLOF variants in *INHBE* (Table 1 and Fig. 3), encoding a member of the activin pathway and transforming growth factor-beta (TGF-β) superfamily known as inhibin βE.

Multiple attributes of this association made it of particular interest. Associations of naturally occurring pLOF alleles with protection from human disease have helped define new therapeutic targets in a growing number of examples[24,25] and this was a newly identified, large-effect association (0.17 SD units; Table 1) with a favorable phenotype (lower BMI-adjusted WHR) for pLOF alleles. Also, in contrast with the exome-wide enrichment for adipose genes, *INHBE* was the only identified gene with strong and specific expression in hepatocytes, but no expression in visceral or subcutaneous adipose tissues (Fig. 4).

The association with favorable fat distribution was consistent in ancestry subsets (Supplementary Data 3) and was strong in men and women with no evidence of sex-interaction (Supplementary Data 9). There was an association with larger hip circumference, but no association with waist (Fig. 5A and Supplementary Data 11). *INHBE* pLOF variants were associated with lower visceral-to-gluteofemoral fat ratio at MRI (beta in SDs of fat ratio per allele, −0.24; 95% CI, −0.45, −0.02;

$p = 0.03$; Supplementary Data 5), and with lower visceral fat volume (Supplementary Data 12). Bioimpedance analyses showed numerically larger impact on body fat rather than lean masses and percentages (Supplementary Fig. 6). *INHBE* pLOF carriers had higher self-reported birthweight and were more likely to self-report a 'plumper-than-average' comparative body size at age 10 (Supplementary Data 13).

We examined the genomic context of the association with BMI-adjusted WHR at the *INHBE* locus and identified no fine-mapped common-variant signals for fat distribution within a 1-Mb window around the gene (Supplementary Fig. 7), consistent with the association being solely driven by rare *INHBE* pLOF alleles. We performed a leave-one-variant-out backward-selection analysis to identify individual rare pLOF alleles contributing to the gene-burden association. The association was primarily but not exclusively driven by a c.299-1 G > C splice acceptor variant accounting for nearly two-thirds of alternative alleles in the aggregate gene-burden genotype (Supplementary Data 14). The c.299-1 G > C variant is in linkage disequilibrium ($r^2 = 0.89$) with a rare Ser544Asn missense variant in *SLC26A10*, a nearby pseudogene tolerant to rare deleterious variation with no reported evidence of protein expression (https://www.proteinatlas.org/)[32]. We performed a number of sensitivity analyses, which supported that the rare-variant signal at the locus is driven by *INHBE* and not *SLC26A10* (Supplementary Result 2), including evidence that: (a) Ser544Asn was not associated with fat distribution after adjusting for c.299-1 G > C; (b) rare coding variants in *INHBE* remained associated with fat distribution even after excluding all Ser544Asn carriers and (c) rare coding variants in *SLC26A10* were not associated with fat distribution after excluding Ser544Asn (Supplementary Result 2, Supplementary Data 10, 11, 14–16). We next expressed the c.299-1 G > C variant in Chinese hamster ovary cells which have no endogenous *INHBE* expression and detected a lower molecular weight protein that was not secreted outside the cell, consistent with loss-of-function (Fig. 6, Supplementary Fig. 8).

Genetic variants associated with favorable fat storage may protect from metabolic disease. In 83,873 cases and 586,592 controls, both the burden of rare pLOF variants (per-allele odds ratio, 0.72; 95% CI, 0.58, 0.90; IVW meta-analysis $p = 0.0043$; Fig. 5B) and the c.299-1G>C splice variant alone (Supplementary Data 11) were associated with lower odds of type 2 diabetes. The association of *INHBE* was similar in magnitude to that of *PDE3B* and *ACVR1C*, two other genes with large-effect associations with favorable fat distribution in our analysis (Supplementary Data 17 and Supplementary Fig. 9) which, similar to *INHBE*, also showed associations with higher hip circumference as a measure of greater gluteofemoral fat (Supplementary Data 8). The association of *INHBE* pLOF variants with protection from diabetes had similar estimates in men and women or in obesity categories in an analysis corrected for potential collider effects (Supplementary Data 18).

A broader exploration of the association of *INHBE* pLOF variants with continuous metabolic traits revealed associations with lower HbA1c, lower apolipoprotein B, lower triglycerides, and higher high-density lipoprotein cholesterol (IVW meta-analysis $p < 0.05$; Fig. 5A and Supplementary Data 11), all of which are consistent with a favorable metabolic phenotype[21,33–35]. There were no associations with estimated bone mineral density or with the risk of bone fracture (Supplementary Result 3).

Given the hepatic expression and proposed role of fat distribution genes in liver dysfunction, we explored associations with liver traits. Rare pLOF variants in *INHBE* were associated with lower alanine transaminase levels (ALT), a measure of liver injury, lower corrected T1 (cT1, an MRI imaging measure of liver inflammation/fibrosis) and lower nonalcoholic fatty liver disease (NAFLD) activity score at liver biopsy in bariatric patients (Supplementary Data 19), though the latter association is driven by only three heterozygous carriers in the bariatric surgery cohort and should be interpreted with caution. We did not observe an association with nonalcoholic liver disease or with liver

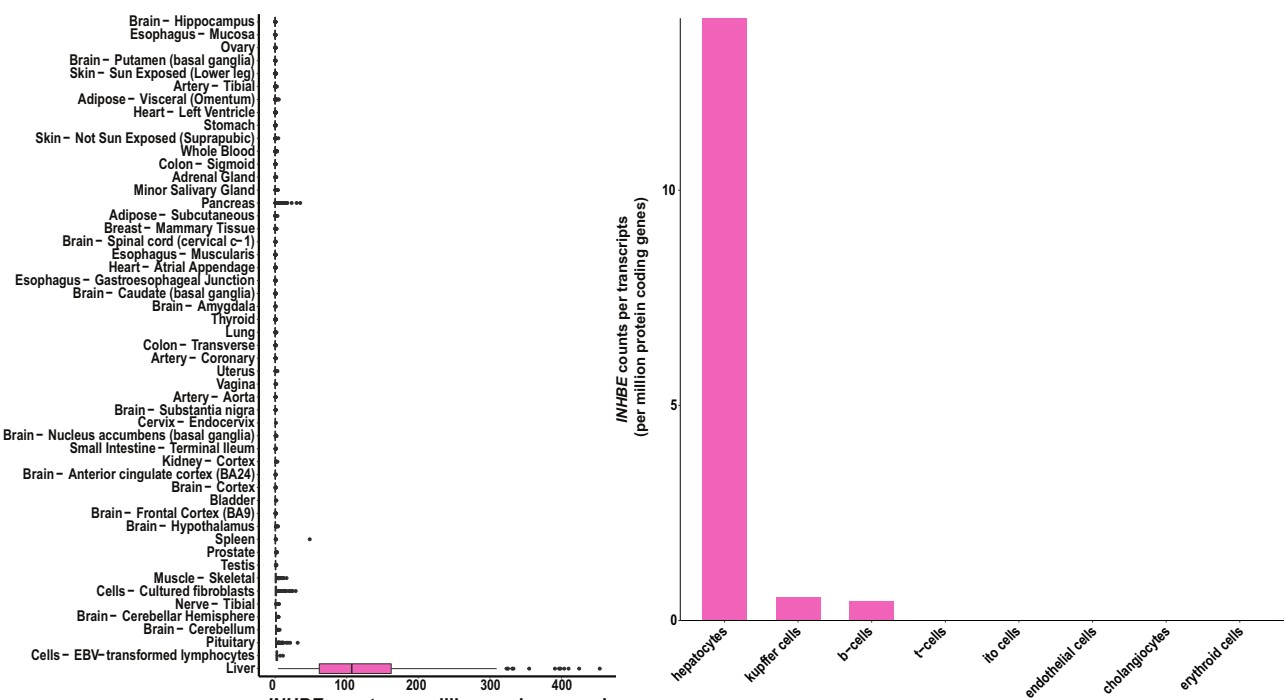

**Fig. 3 | Protein-truncating variants in *INHBE* associated with favorable fat distribution.** The top panel is a linear model of the *INHBE* protein with prodomain and mature domains in gray and brown, respectively. An exon track shows the two exons of the gene (in purple and blue). Predicted loss of function (pLOF) variants identified by exome-sequencing and included in the gene-burden analysis are shown below the exon track, with numbers in parenthesis corresponding to the number of carriers for each allele. The bottom panel shows associations with fat distribution for pLOF variants in *INHBE* across cohorts. P-values are from two-sided Wald tests. Markers represent the beta estimates while error bars represent 95%

confidence intervals. The gray diamond represents meta-analysis estimates. BMI body mass index, WHR waist-hip ratio, pLOF predicted loss of function, AAF alternative allele frequency, UKB UK Biobank, MDCS Malmö Diet and Cancer Study, MCPS Mexico City Prospective Study, AFR African ancestry, SAS south-Asian ancestry, EUR European ancestry, AMR admixed-American ancestry, RR reference-reference homozygous genotype, RA reference-alternative heterozygous genotype, AA alternative-alternative homozygous genotype, CI confidence interval, SD standard deviations, *P* P-value.

**Fig. 4 | *INHBE* mRNA expression in humans across tissues and liver cell-types.** The left panel shows normalized mRNA expression for *INHBE* in counts per million (CPM) across tissues from the Genotype Tissue Expression (GTEx) consortium[107], box plots depict the median, interquartile range, and range of CPM values across

individuals for each tissue[107]. The right panel shows normalized cell-type specific expression within liver in counts per transcripts per million protein coding genes (pTPM) from the Human Protein Atlas (HPA)[108].

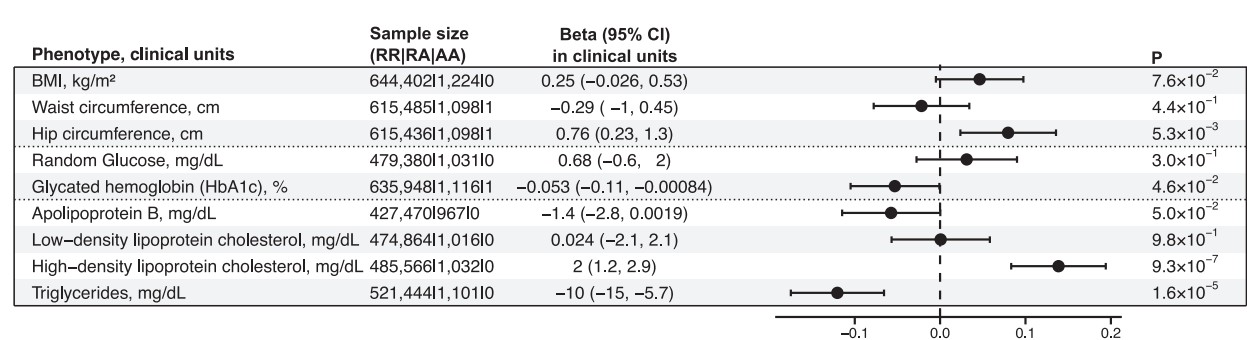

**A**

| Phenotype, clinical units | Sample size (RR\|RA\|AA) | Beta (95% CI) in clinical units | | P |
|---|---|---|---|---|
| BMI, kg/m² | 644,402\|1,224\|0 | 0.25 (−0.026, 0.53) | | 7.6×10⁻² |
| Waist circumference, cm | 615,485\|1,098\|1 | −0.29 ( −1, 0.45) | | 4.4×10⁻¹ |
| Hip circumference, cm | 615,436\|1,098\|1 | 0.76 (0.23, 1.3) | | 5.3×10⁻³ |
| Random Glucose, mg/dL | 479,380\|1,031\|0 | 0.68 (−0.6, 2) | | 3.0×10⁻¹ |
| Glycated hemoglobin (HbA1c), % | 635,948\|1,116\|1 | −0.053 (−0.11, −0.00084) | | 4.6×10⁻² |
| Apolipoprotein B, mg/dL | 427,470\|967\|0 | −1.4 (−2.8, 0.0019) | | 5.0×10⁻² |
| Low−density lipoprotein cholesterol, mg/dL | 474,864\|1,016\|0 | 0.024 (−2.1, 2.1) | | 9.8×10⁻¹ |
| High−density lipoprotein cholesterol, mg/dL | 485,566\|1,032\|0 | 2 (1.2, 2.9) | | 9.3×10⁻⁷ |
| Triglycerides, mg/dL | 521,444\|1,101\|0 | −10 (−15, −5.7) | | 1.6×10⁻⁵ |

Beta (95% CI) in SD units per *INHBE* pLOF allele

**B**

**Type 2 diabetes (INHBE − pLoF, AAF < 1%)**

| Cohort | Ancestry | Cases RR\|RA\|AA | Controls RR\|RA\|AA | Odds ratio (95% CI) | | P |
|---|---|---|---|---|---|---|
| UKB | SAS | 1,936\|0\|0 | 8,195\|5\|0 | 0.29 (0.02, 3.82) | | 3.5 × 10⁻¹ |
| SINAI | EUR | 978\|1\|0 | 7,492\|6\|0 | 1.18 (0.11, 12.24) | | 8.9 × 10⁻¹ |
| MDCS | EUR | 3,802\|2\|0 | 21,117\|22\|0 | 0.69 (0.20, 2.38) | | 5.6 × 10⁻¹ |
| MCPS | AMR | 26,482\|6\|1 | 81,936\|43\|0 | 0.71 (0.34, 1.48) | | 3.6 × 10⁻¹ |
| GHS | EUR | 26,740\|18\|0 | 64,737\|111\|0 | 0.49 (0.30, 0.80) | | 4.1 × 10⁻³ |
| UKB | EUR | 23,862\|45\|0 | 401,975\|953\|0 | 0.82 (0.62, 1.09) | | 1.7 × 10⁻¹ |
| **Meta−analysis** | **ALL** | **83,800\|72\|1** | **585,452\|1,140\|0** | **0.72 (0.58, 0.90)** | | 4.3 × 10⁻³ |

Heterogeneity I2=0%; P=0.55

Odds ratio (95% CI)

**Fig. 5 | Association of *INHBE* pLOF variants with favorable metabolic profile and protection from type 2 diabetes. A** shows associations with anthropometric and metabolic phenotypes, including 645,626 individuals. *P*-values are from two-sided Wald tests. Markers represent estimated beta coefficients, while error bars represent 95% confidence intervals. **B** shows a meta-analysis of the association with type 2 diabetes risk, including a total of 83,873 cases and 586,592 controls. P-values are from two-sided Wald tests. Markers represent estimated odds ratios, while error bars represent 95% confidence intervals. The gray diamond represents meta-analysis estimates. AAF alternative allele frequency; BMI body mass index; RR reference-reference homozygous genotype; RA reference-alternative heterozygous genotype; AA alternative-alternative homozygous genotype; CI confidence intervals; *P* P-value; SD standard deviation; pLOF predicted loss of function; kg kilogram; m² meter squared; mg milligram; dL deciliter; cm centimeter; UKB UK Biobank study; SINAI Mount Sinai BioMe cohort; MDCS Malmö Diet and Cancer Study; MCPS Mexico City Prospective Study; GHS Geisinger Health System; EUR European; SAS South Asian; AMR American; ALL all ancestries pooled.

cirrhosis outcomes, but the analysis was underpowered due to the rarity of *INHBE* pLOF alleles (Supplementary Data 19).

We performed RNASeq in liver biopsy samples from a cohort of bariatric surgery patients ("Methods") and investigated the association between liver disease status and *INHBE* expression. Individuals with liver steatosis exhibited higher *INHBE* expression compared to individuals with healthy liver (25% higher expression; Wald test $p = 4.1 \times 10^{-16}$; Supplementary Fig. 10), while individuals with nonalcoholic steatohepatitis had even higher expression (60% higher compared to healthy liver; Wald test $p = 2.0 \times 10^{-63}$; Supplementary Fig. 10). Furthermore, we observed a strong association between higher NAFLD activity score at liver biopsy and higher liver expression of *INHBE* mRNA (Supplementary Fig. 10). *INHBE* hepatic expression showed modest correlation with that of activin A or follistatin (Supplementary Fig. 11), which are other members of the TGF-β family involved in metabolic regulation and disease[36,37].

Overall, our results suggest that inhibin βE is a liver-derived negative regulator of energy storage in peripheral adipose tissue in humans and that its inactivation may protect from metabolic disease.

### Genetic evidence of a central role for liver steatosis and inflammation in fat distribution-associated disease

Hepatic steatosis and inflammation have been proposed to play a central role in fat distribution related cardiometabolic disease[1], but genetic evidence of this mechanism is lacking. Here, we studied associations with (a) transaminase levels in 542,904 people; (b) MRI-derived measures of liver fat (proton-density fat fraction, PDFF) and liver inflammation/fibrosis (cT1) in 36,402 people; (c) liver histopathology in 3565 bariatric surgery patients; and (d) liver disease diagnoses in 15,851 cases and 468,511 controls. In observational epidemiology analyses, higher BMI-adjusted WHR and higher BMI showed the expected association with higher levels of MRI-measured liver fat, inflammation and with higher risk of liver disease outcomes and type 2 diabetes (Supplementary Figs. 12 and 13).

We next estimated associations with liver phenotypes for four validated[16,21] common-variant scores capturing polygenic predisposition to: (a) lower WHR via both higher gluteofemoral and lower abdominal fat; (b) lower WHR via lower abdominal fat (waist-specific score) (c) lower WHR via higher gluteofemoral fat (hip-specific score); and (d) lower insulin resistance via greater adipose expandability. We used a polygenic score for lower BMI as comparator [38].

The favorable fat distribution scores were associated with "gold-standard" measures of adipose expandability, peripheral adiposity and fat distribution at dual-energy X-ray absorptiometry (DXA) in a small subset of UKB (*N* = 5117 or 0.8% of the discovery sample with available DXA; Supplementary Fig. 14). Favorable fat distribution polygenic scores were strongly associated with lower transaminase levels, lower liver fat and inflammation at MRI (Fig. 7A, Supplementary Fig. 15), as well as protection from nonalcoholic liver disease and liver cirrhosis (Fig. 7A and Supplementary Fig. 15). The polygenic score for lower

A

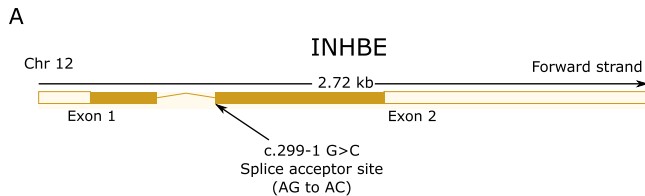

B

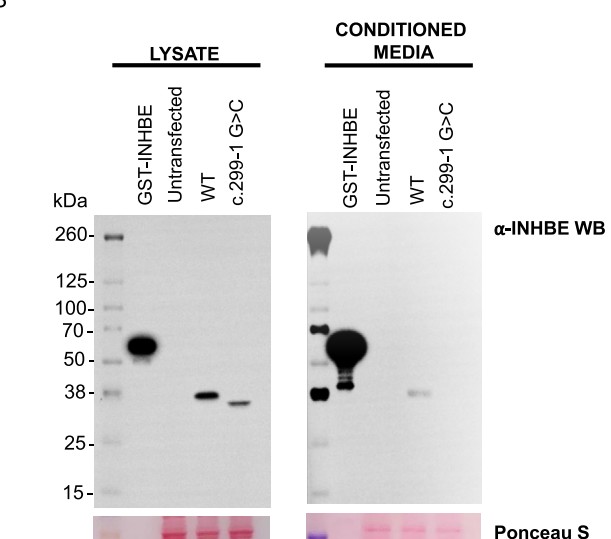

**Fig. 6 | In vitro expression of the *INHBE* c.299-1G>C splice acceptor variant.**
**A** shows a gene model for *INHBE* with the predicted loss of function splice acceptor variant (c.299-1G>C) highlighted. **B** shows a Western blot analysis for INHBE protein in cell lysates and conditioned media from CHO cells transfected with wild-type *INHBE* or the *INHBE* c.299-1G>C splice acceptor variant. Full length GST-tagged recombinant INHBE protein (100 ng) was used as a positive control and staining with Ponceau S was used to compare sample loading across samples. The image is representative of one of three technical replicates. Each replicate yielded similar results. Chr chromosome, CHO Chinese hamster ovary, GST Glutathione s-transferase, WT wild type, kDa kilodalton, WB western blot, Ponceau S Ponceau S (Acid Red 112).

insulin resistance via greater adipose expandability and the overall score for lower BMI-adjusted WHR were also associated with lower NAFLD activity score on histopathology (Fig. 7A and Supplementary Fig. 15). Notably, associations of fat distribution polygenic scores with liver traits were similarly strong and, at times, even stronger than those of the polygenic score for lower BMI, for a given genetically determined difference in the underlying trait. All four scores were also robustly associated with lower risk of type 2 diabetes and coronary artery disease (Fig. 7A and Supplementary Fig. 15), consistent with previous literature[15,16,21], and associations with diabetes and liver disease were independent of known epidemiologic risk factors for these conditions (Supplementary Data 20).

Next, we investigated relationships with liver and cardiometabolic disease outcomes for the genes identified in our gene-burden analysis of fat distribution, pooling associations across 16 genes using a rare-variant Mendelian randomization approach to maximize statistical power. Genetically lower BMI-adjusted WHR via the 16 genes was associated with lower ALT, lower liver fat and lower risk of type 2 diabetes and coronary disease (Fig. 7B), consistent with the common-variant associations. Statistically significant associations with type 2 diabetes showed no evidence of sex-interaction, while the association with coronary disease appeared stronger in men (Supplementary Data 21). As fat distribution is associated with metabolic disease also in lean individuals[39], we estimated associations with type 2 diabetes in a BMI-stratified analysis which accounts for possible collider bias and

found that associations were similarly strong by obesity category (Supplementary Data 22).

## Interplay of rare and common fat distribution variants in the general population

Given the observation that polygenic extremes and Mendelian-disease causing variants have a similar impact on certain traits[40,41], the combined availability of exome-sequencing, genome-wide genotyping and fat distribution phenotypes in our study, and the observed enrichment for FPLD-causing genes in the exome-wide analysis, we investigated the interplay of common and rare variants for fat distribution.

We compared associations for mutations in *PPARG*, the causal gene for FPLD type 3 (used as benchmark for Mendelian-like effects; "Methods"), with those of a genome-wide polygenic score for BMI-adjusted WHR generated and validated using GWAS data from our analysis ("Methods"; Supplementary Fig. 16). *PPARG* mutation carriers had 0.46 SD higher BMI-adjusted WHR (IVW meta-analysis $p = 0.012$; Table 2, Supplementary Data 23) and >4-fold higher odds of type 2 diabetes (IVW meta-analysis $p = 3.4 \times 10^{-4}$; Table 2, Supplementary Data 23) compared to noncarriers; which is consistent with the effect-size of other Mendelian mutations in population-based studies[22,40,41]. In the same dataset, the genome-wide polygenic score was robustly associated with fat distribution and related-disease (Supplementary Result 4, Supplementary Figs. 17–20, Supplementary Data 24, 25), with individuals in the top 1% of polygenic predisposition having similar average fat distribution as *PPARG* mutation carriers (0.58 SDs; Table 2). Notably, being in the top 1% of the polygenic score is approximately 120-times more frequent than being the heterozygous carrier of a *PPARG* mutation in the cohorts we studied (Table 2). Other genotype combinations including rare alleles combined with high polygenic burden or multiple rare alleles for other genes identified in our exome-wide analysis had similar impact (*ANKRD12*, *PLIN4*; Supplementary Result 4; Supplementary Data 24). At the opposite polygenic extreme, individuals in the bottom 1% of the polygenic score had a favorable fat distribution (−0.67 SDs; Table 2) and a risk of type 2 diabetes similar to that of *INHBE* pLOF carriers (Table 2).

## Discussion

We performed a large and ancestrally diverse study on the influences of rare coding variants on body shape and associated cardiometabolic disease, making a number of observations that substantially advance our understanding of the genetic basis of these phenotypes.

First, we showed that rare mutations in numerous genes have a substantial impact on body fat distribution in the general population. We identified new associations with favorable adiposity and protection from metabolic disease for rare loss-of-function variants in *INHBE*, encoding a liver-produced circulating member of the TGF-β superfamily. Our results suggest that inhibin βE is a liver-expressed negative regulator of energy storage in peripheral adipose tissue in humans and that loss of its function protects from liver inflammation, dyslipidemia and type 2 diabetes by promoting healthy fat storage. These findings may have therapeutic implications. The identification of naturally ccurring loss-of-function variants associated with protection from human disease has helped identify a growing number of new targets for pharmacological inhibition across multiple indications[22–25,42,43]. Human genetic support is associated with higher odds of successful drug development[44] and hepatocyte-expressed genes that encode circulating proteins like *INHBE* can be effectively inhibited via liver-directed oligonucleotide therapeutics or by monoclonal antibodies targeting the circulating protein product, as shown in several clinical trials[45–51]. Hence, inhibition of INHBE may be a therapeutic approach for metabolic disease associated with improper fat storage.

*INHBE* provides an example of a liver-specific gene where rare loss-of-function mutations are associated with body fat distribution.

## A  Common variants associations

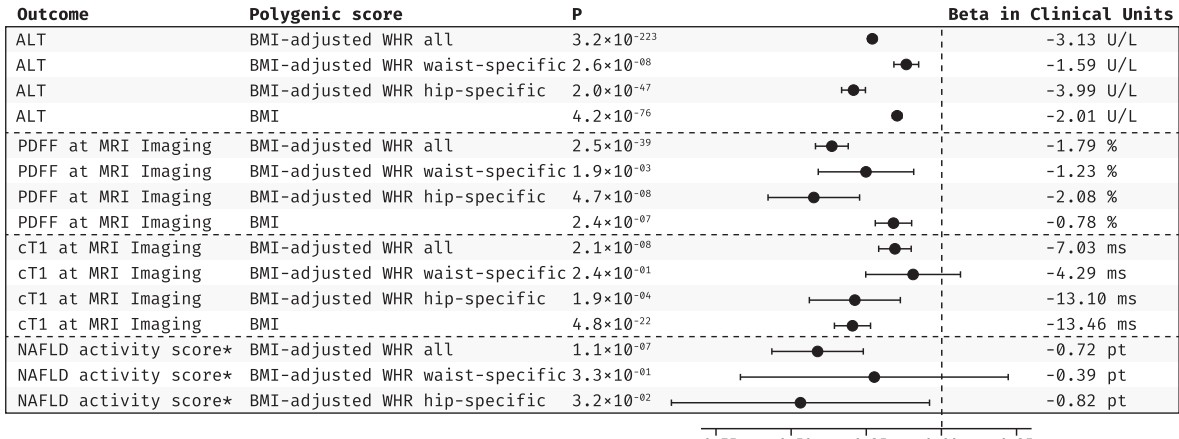

| Outcome | Polygenic score | P | | Beta in Clinical Units |
|---|---|---|---|---|
| ALT | BMI-adjusted WHR all | $3.2\times10^{-223}$ | | -3.13 U/L |
| ALT | BMI-adjusted WHR waist-specific | $2.6\times10^{-08}$ | | -1.59 U/L |
| ALT | BMI-adjusted WHR hip-specific | $2.0\times10^{-47}$ | | -3.99 U/L |
| ALT | BMI | $4.2\times10^{-76}$ | | -2.01 U/L |
| PDFF at MRI Imaging | BMI-adjusted WHR all | $2.5\times10^{-39}$ | | -1.79 % |
| PDFF at MRI Imaging | BMI-adjusted WHR waist-specific | $1.9\times10^{-03}$ | | -1.23 % |
| PDFF at MRI Imaging | BMI-adjusted WHR hip-specific | $4.7\times10^{-08}$ | | -2.08 % |
| PDFF at MRI Imaging | BMI | $2.4\times10^{-07}$ | | -0.78 % |
| cT1 at MRI Imaging | BMI-adjusted WHR all | $2.1\times10^{-08}$ | | -7.03 ms |
| cT1 at MRI Imaging | BMI-adjusted WHR waist-specific | $2.4\times10^{-01}$ | | -4.29 ms |
| cT1 at MRI Imaging | BMI-adjusted WHR hip-specific | $1.9\times10^{-04}$ | | -13.10 ms |
| cT1 at MRI Imaging | BMI | $4.8\times10^{-22}$ | | -13.46 ms |
| NAFLD activity score* | BMI-adjusted WHR all | $1.1\times10^{-07}$ | | -0.72 pt |
| NAFLD activity score* | BMI-adjusted WHR waist-specific | $3.3\times10^{-01}$ | | -0.39 pt |
| NAFLD activity score* | BMI-adjusted WHR hip-specific | $3.2\times10^{-02}$ | | -0.82 pt |

Beta (95% CI) per 1 SD lower polygenic score

| Outcome | Polygenic score | P | | Odds ratio |
|---|---|---|---|---|
| Nonalcoholic liver disease | BMI-adjusted WHR all | $5.5\times10^{-09}$ | | 0.74 |
| Nonalcoholic liver disease | BMI-adjusted WHR waist-specific | $7.4\times10^{-03}$ | | 0.67 |
| Nonalcoholic liver disease | BMI-adjusted WHR hip-specific | $1.0\times10^{-04}$ | | 0.57 |
| Nonalcoholic liver disease | BMI | $5.3\times10^{-45}$ | | 0.46 |
| Cirrhosis | BMI-adjusted WHR all | $1.8\times10^{-04}$ | | 0.71 |
| Cirrhosis | BMI-adjusted WHR waist-specific | $7.5\times10^{-01}$ | | 0.92 |
| Cirrhosis | BMI-adjusted WHR hip-specific | $3.3\times10^{-05}$ | | 0.34 |
| Cirrhosis | BMI | $9.9\times10^{-06}$ | | 0.64 |
| Type 2 Diabetes | BMI-adjusted WHR all | $3.9\times10^{-132}$ | | 0.51 |
| Type 2 Diabetes | BMI-adjusted WHR waist-specific | $1.2\times10^{-07}$ | | 0.66 |
| Type 2 Diabetes | BMI-adjusted WHR hip-specific | $1.4\times10^{-42}$ | | 0.35 |
| Type 2 Diabetes | BMI | $9.8\times10^{-176}$ | | 0.43 |
| Coronary Artery Disease | BMI-adjusted WHR all | $1.7\times10^{-38}$ | | 0.73 |
| Coronary Artery Disease | BMI-adjusted WHR waist-specific | $1.4\times10^{-10}$ | | 0.64 |
| Coronary Artery Disease | BMI-adjusted WHR hip-specific | $3.0\times10^{-19}$ | | 0.55 |
| Coronary Artery Disease | BMI | $3.9\times10^{-29}$ | | 0.74 |

Odds ratio (95% CI) per 1 SD lower polygenic score

## B  Rare variants associations

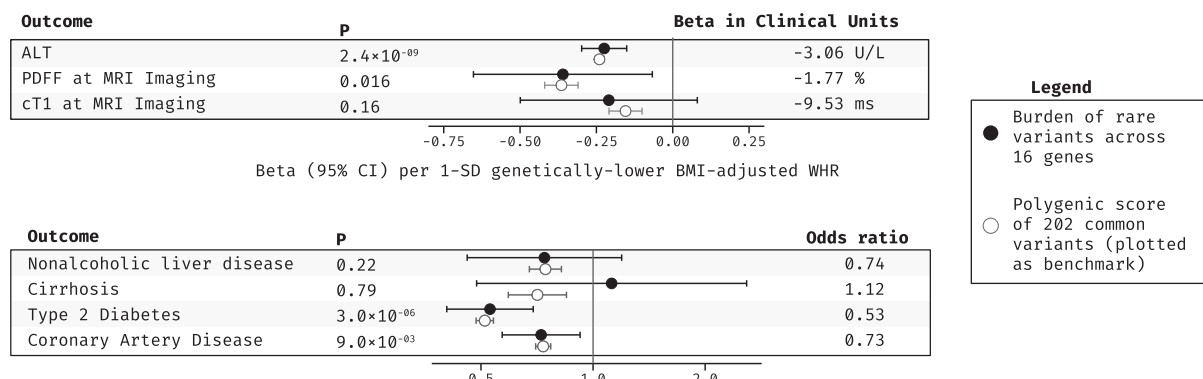

| Outcome | P | | Beta in Clinical Units |
|---|---|---|---|
| ALT | $2.4\times10^{-09}$ | | -3.06 U/L |
| PDFF at MRI Imaging | 0.016 | | -1.77 % |
| cT1 at MRI Imaging | 0.16 | | -9.53 ms |

Beta (95% CI) per 1-SD genetically-lower BMI-adjusted WHR

**Legend**

● Burden of rare variants across 16 genes

○ Polygenic score of 202 common variants (plotted as benchmark)

| Outcome | P | | Odds ratio |
|---|---|---|---|
| Nonalcoholic liver disease | 0.22 | | 0.74 |
| Cirrhosis | 0.79 | | 1.12 |
| Type 2 Diabetes | $3.0\times10^{-06}$ | | 0.53 |
| Coronary Artery Disease | $9.0\times10^{-03}$ | | 0.73 |

OR (95% CI) per 1-SD genetically-lower BMI-adjusted WHR

This association uncovers a potential new player in the biological interactions between liver, a critical organ for energy sensing, and adipose tissue, the specialized energy storage system of the human body. Higher levels of *INHBE* expression have been observed in insulin resistance[52], an early pathophysiologic process in metabolic disease, and, in our study, in hepatic steatosis and inflammation, which may partly reflect the insulin resistance associated with those conditions.

We hypothesize that the upregulation of *INHBE* in those settings may drive a maladaptive response to excess calories.

The biological functions of inhibin βE in humans are largely unknown and could be disparate, given its potential to form homodimers or heterodimers with other members of the activin family which are involved in multiple processes[53]. Interestingly, inhibin βE has recently been proposed to be a hepatokine that regulates energy

**Fig. 7 | Genetic associations of favorable fat distribution with liver phenotypes, type 2 diabetes and coronary artery disease risk. A** shows associations for common variants polygenic scores for favorable fat distribution; associations for a BMI polygenic score are shown for comparison. Markers and error bars represent beta coefficients (for continuous traits) or odds ratios (for binary traits) and their 95% confidence intervals. *P*-values are from two-sided Wald tests. Sample sizes: ALT, 442,695; PDFF at MRI imaging, 38,915; cT1 at MRI imaging, 38,915; NAFLD activity score, 3572; nonalcoholic liver disease, 14,195 cases and 428,139 controls; cirrhosis, 4063 cases and 428,139 controls; type 2 diabetes, 58,379 cases and 530,072 controls; coronary artery disease, 89,202 cases and 342,007 controls; *NAFLD activity score measured by liver biopsy, the association for the BMI polygenic score was not estimated for this phenotype as it is only available in bariatric surgery patients with extremely high BMI. **B** shows associations for rare coding

variants (full black circles) and a common-variant polygenic score for lower BMI-adjusted WHR (open circles; shown as benchmark). Markers and error bars represent beta coefficients (for continuous traits) or odds ratios (for binary traits) and their 95% confidence intervals. P-values are from two-sided Wald tests. P-values are from two-sided Wald tests. ALT, 542,904; PDFF at MRI imaging, 37,686; cT1 at MRI imaging, 37,686; NAFLD activity score, 3565; nonalcoholic liver disease, 15,858 cases and 468,523 controls; cirrhosis, 4950 cases and 466,464 controls; types 2 diabetes, 66,062 cases and 530,538 controls; coronary artery disease, 92,824 cases and 361,297 controls. ALT alanine aminotransferase, PDFF proton-density liver fat fraction, MRI magnetic resonance imaging, cT1 corrected T1, NAFLD nonalcoholic fatty liver disease, BMI body mass index, WHR waist-hip ratio, P P-value, CI confidence intervals, SD standard deviation, U/L units per liter, ms milliseconds, pt point.

homeostasis by inducing beige adipocytes and improving insulin sensitivity in mouse models of hepatic overexpression[54]. While the role as liver-produced regulator of adipose function is consistent with our human genetic findings, the directionality of associations is opposite. In the mouse model, overexpression of inhibin βE resulted in greater insulin sensitivity[54], whereas loss-of-function is associated with protection from metabolic disease in humans. Notably, different mouse models of inhibin βE perturbation have yielded contrasting results[52,54]. More broadly, phenotypic inconsistencies have been highlighted between mutations affecting fat distribution in humans and mouse models of those variants[55,56], which are partly due to inter-species differences in adipose patterning and function. Therefore, the relevance of mouse models of inhibin βE perturbation to human pathophysiology is unclear.

Second, by using genomic data in conjunction with "gold-standard" liver MRI imaging and histopathology phenotypes, our study shows the profound impact of adipose expandability genes on ectopic liver fat deposition, hepatic inflammation and disease, uncovering another key aspect of adipose-liver interplay in energy storage. We show that variation at adipose-expressed genes associated with an enhanced ability to expand peripheral fat storage results in lower levels of liver fat, lower liver inflammation, and protection against cirrhosis. Our results show that improper adipose storage is a key determinant of the burgeoning epidemic of nonalcoholic liver disease and suggest that enhancing adipose expandability may be an important preventive or therapeutic strategy for these conditions.

Third, our results highlight the combined impact of common and rare variation on fat distribution in the general population. Individually, rare coding genotypes had much larger phenotypic impact than that of common alleles. However, the cumulative impact of common polygenic predisposition (captured by polygenic score extremes) was as large as that of *PPARG* mutations, with an over 100-fold higher frequency. These results illustrate the existence of more prevalent polygenic forms of lipodystrophy-like disease, and may help explain the observation that mutations in known causal genes cannot be identified in a large proportion of patients with FPLD [21,57].

This study has limitations. The coupling of exome-sequencing at scale in diverse ancestries and the confidence in effector gene attribution afforded by rare coding variants enabled us to pinpoint several effector genes for fat distribution. However, the rarity of some of the associated alleles means that the number of associated loci for a given sample size is higher for GWAS of common variants, and suggests that sequencing of millions of people across ancestries and geographies will be necessary to fully catalogue the impact of rare variation on these traits and the contribution of individual alleles in identified genes. Also, WHR is a simple and broadly used proxy-measure of fat distribution, but does not fully capture the spectrum of variation in human body composition. Here, we used "gold-standard" MRI-based measures of fat distribution and several sensitivity analyses to validate the identified associations. Human genetic analyses centered on refined imaging phenotypes may reveal more detailed patterns and insights.

In summary, this study identified genes where rare coding alleles are associated with large differences in body fat distribution in humans, including an association with protection against metabolic disease for rare loss-of-function variants in the liver-expressed *INHBE*. Our results suggest that blocking inhibin βE may be a therapeutic approach for promoting metabolic health and uncover biological interplays between liver and adipose tissue in energy storage.

## Methods
### Participating cohorts
Exome-wide analyses were performed in UK Biobank (UKB)[58], Malmö Diet and Cancer study (MDCS)[59], and Mexico City Prospective Study (MCPS)[60]. UKB is a population-based cohort of people 40-69 years of age recruited in the UK in 2006-2010. A total of 429,442 European, 10,115 South Asian, 8,948 African, 2,203 East Asian, 604 American ancestry participants with exome sequencing and phenotypic data were included (Supplementary Data 1). MDCS is a population-based cohort of 44–73-year-old people living in Malmö (Sweden) and recruited in 1991–1996. A total of 28,875 European ancestry participants were included (Supplementary Data 1). MCPS is a population-

## Table 2 | Fat distribution and prevalence of type 2 diabetes in high-impact genetic exposure groups

| Variable | Overall population | *PPARG* mutations[a] | Top 1% of BMI-adjusted WHR polygenic score | *INHBE* pLOF variants | Bottom 1% of BMI-adjusted WHR polygenic score |
|---|---|---|---|---|---|
| Frequency for each exposure group (number of people needed to sequence to observe one individual in the genetic exposure group) | - | 0.0083% (1 in 12,048) | 1% (1 in 100) | 0.22% (1 in 454) | 1% (1 in 100) |
| Mean BMI-adjusted WHR in each exposure group (95% CI of the mean) in SD units of BMI-adjusted WHR | 0 (−0.002, 0.004) | 0.46 (0.05, 0.86) | 0.58 (0.56, 0.61) | −0.19 (−0.25, −0.13) | −0.67 (−0.70, −0.64) |
| Prevalence of type 2 diabetes in each exposure group (95% CI), percentage | 9.8% (9.7%, 9.9%) | 33% (19%, 47%) | 13% (12%, 14%) | 5.7% (4.3%, 7.1%) | 6.9% (6.2%, 7.6%) |

Estimates are from European ancestry participants in UKB (for BMI-adjusted WHR; *n* = 428,652) and UKB plus GHS (for diabetes; 50,167 cases and 466,291 controls).
*pLOF* predicted loss of function, *BMI* body mass index, *WHR* waist-to-hip ratio, *SD* standard deviation, *CI* confidence interval.
[a]Genotype includes rare protein-truncating variants or experimentally validated loss-of-function variants predicted to be causal for FPLD type 3 ("Methods").

based cohort of people aged 35 years or older, recruited from two urban districts in Mexico City in 1998–2004[60,61]. A total of 138,188 participants of American ancestry were included (Supplementary Data 1). Ancillary analyses included association results from 109,909 participants in the Geisinger Health System MyCode and DiscovEHR collaborations (GHS)[62,63], 28,338 participants in the Mount Sinai BioMe biobank cohort (BioMe; mean age, 55 years; 59% women)[64], and 15,046 participants in the University of Pennsylvania PennMedicine Biobank cohort (mean age, 63 years; 52% women)[65]. Ethical approval for the UKB was obtained from the North West Centre for Research Ethics Committee (11/NW/0382) and the work described here was approved by UK Biobank under application number 26041. The MCPS study was approved by the Mexican Ministry of Health, the Mexican National Council for Science and Technology, and the University of Oxford. The MDCS study was approved by the Regional Ethics Committee at Lund University. Approval for DiscovEHR analyses was provided by the Geisinger Health System Institutional Review Board under project number 2006-0258. Approval for the University of Pennsylvania Penn Medicine Biobank was provided by the Institutional Review Board of the University of Pennsylvania. Mount Sinai BioMe biobank cohort was approved by the Icahn School of Medicine at Mount Sinai's Institutional Review Board.

## Phenotype definitions

Our primary trait of interest was BMI-adjusted WHR, a phenotype which has been widely used in human genetic studies of fat distribution[14,16,18,19]. BMI-adjusted WHR was defined as the ratio between waist and hip circumference adjusted for BMI, calculated as weight in kilograms divided by the square of height in meters, as previously done[14,16,18,19]. Adjustment for BMI was performed by calculation of residuals in a linear regression model with WHR as outcome and BMI as exposure. The inverse-rank normal transformation was then applied in sex- and ancestry-specific subgroups.

Blood biomarkers were analyzed in UKB, GHS and MCPS. In UKB, HbA1c was analyzed using high-performance liquid chromatography (VARIANT II Turbo Hemoglobin Testing System, Bio-Rad), and glucose, liver enzymes (alanine aminotransferase [ALT] and aspartate aminotransferase [AST]), and blood lipids (apolipoprotein B, triglycerides, high-density lipoprotein cholesterol, and low-density lipoprotein cholesterol) were analyzed using the AU5800 clinical chemistry analyzer (Beckman Coulter). In GHS, biomarker values were extracted from the electronic medical records, as described previously[24]. In MCPS, HbA1c was analyzed using high-performance liquid chromatography (HA-8180 analyzers, Arkray).

Type 2 diabetes cases were adjudicated in each cohort based on one or more of the following criteria: (1) an electronic health record of type 2 diabetes (using International Classification of Diseases, Tenth Revision [ICD-10] diagnosis codes E11 or O24.1 or corresponding Ninth Revision [ICD-9] codes), in at least one inpatient encounter or at least two outpatient encounters or if noted as a cause of death; (2) a glycemic biomarker value (HbA1c, random or fasting glucose) in the diabetic range[66]; (3) a prescription record of anti-diabetic medication use; (4) a self-reported physician diagnosis of type 2 diabetes; (5) entry on a diabetes registry as a type 2 diabetes case. Where possible, we excluded individuals from the case pool if they had a potential diagnosis of type 1 diabetes mellitus (using ICD-10 codes E10 or O24.0, or a prescription record that included insulin only in the absence of other diabetic medication). Individuals not meeting any of the criteria for diabetes case status were used as controls. In addition, individuals were excluded from the control group if they met any of the following criteria: (1) an electronic health record diagnosis pertaining to any potential type of diabetes mellitus or a family history of diabetes; (2) a glycemic biomarker value in the prediabetic range[66]; (3) any other cohort-specific phenotype that potentially indicated a diagnosis of diabetes mellitus (e.g. a disease registry entry or self-reported diagnosis of non-specific diabetes).

Liver disease (nonalcoholic liver disease and liver cirrhosis) cases were defined using one or more of the following criteria: (1) an electronic health record of disease, in at least one inpatient encounter, or at least two outpatient encounters, or if noted as a cause of death; (2) self-reported disease, ascertained at study recruitment; (3) surgery or medical procedures performed for the disease. Individuals not meeting any of the case criteria were used as controls. Subjects were also excluded from the control group if they met any of the following: (1) diagnosis of any type of liver disease (i.e. beyond NAFLD or liver cirrhosis); (2) presence of only a single outpatient encounter for the liver disease of interest; (3) had elevated ALT (>25 IU/L for women and >33 IU/L for men[67]); (4) had a diagnosis of ascites attributed to liver disease. Diagnostic codes used for liver diseases are shown in Supplementary Data 26. Coronary artery disease (CAD) cases were defined using one or more of the following criteria: (1) an electronic health record of CAD and/or myocardial infarction, in at least one inpatient encounter, or at least two outpatient encounters, or if noted as a cause of death; (2) self-reported CAD or myocardial infarction, ascertained at study recruitment; (3) surgery or medical procedures performed for CAD, including coronary artery bypass grafting and/or percutaneous coronary intervention. We further excluded individuals with a family history of CAD (defined using EHR diagnostic codes or self-reported data) from the control group. Fracture was defined as a history of electronic health record-coded or self-reported vertebral or non-vertebral fracture (the latter not including fractures of the skull, facial bones, hands, or toes, where possible). We excluded individuals with a history of any type of fracture from the control group.

## Liver histopathology and MRI phenotypes

Liver histopathology phenotypes were derived in 3,565 European-ancestry individuals who underwent bariatric surgery and were enrolled in the GHS-RGC DiscovEHR collaboration[62]. Liver histology was assessed on intraoperative wedge biopsies of the liver by an experienced histopathologist and reviewed by a second pathologist. All biopsies were scored using the NASH Clinical Research Network system[68].

A subset of ~36,000 participants in UKB underwent magnetic resonance imaging (MRI) of the liver, using Siemens MAGNETOM Aera 1.5T clinical MRI scanners[69]. This included two liver acquisitions: a quantitative T1 mapping sequence and a sequence for estimating liver fat content. For T1 mapping, a "ShMOLLI" (Shortened Modified Look-Locker Inversion recovery) protocol was used. Since T1 measurements may be confounded by liver iron levels, we derived iron-corrected T1 (cT1) values as described[70]. Higher cT1 values correlate with liver inflammation and fibrosis on histology[70,71]. For liver fat imaging, the first ~10,000 participants (pre-2016) were imaged using a Dixon gradient echo protocol, whilst all further participants were imaged using the IDEAL (Iterative Decomposition of water and fat with Echo Asymmetry and Least-squares estimation) sequence. We derived measurements of proton-density liver fat fraction (PDFF, estimated as the fraction of fat signal relative to total fat and water signal) by applying pre-defined mathematical models after segmenting the liver images[72–74]. We used an automated workflow to segment pixels belonging to the liver using a Li thresholding approach for PDFF maps. All liver pixels were subsequently averaged for each parametric map, to obtain a measure of each trait. Full details of these approaches have been previously described elsewhere[75].

## Gold-standard measures of fat distribution

A subset of ~46,000 participants in UKB underwent two-point Dixon[76] MRI using Siemens MAGNETOM Aera 1.5 T clinical MRI scanners[69], split into six different imaging series. This subset included 38,880 people with available exome sequencing. Stitching of the six different scan

positions corrected for overlapping slices, partial scans, repeat scans, fat-water swaps, misalignment between imaging series, bias-field, artificially dark slices and local hotspots, similar to what has previously been performed[77]. A total of 52 subjects had their whole-body Dixon MRI manually annotated into six different classes of fat: upper body fat, abdominal fat, visceral fat, mediastinal fat, gluteofemoral fat and lower-leg fat. These annotations were then used to train a multi-class segmentation deep neural net which employed a UNet[78] architecture with a ResNet34[79] backbone, and a loss function of a sum of the Jaccard Index and categorical focal loss[80]. Fat volume phenotypes were calculated by summing the resulting segmentation maps from the neural net for each corresponding fat class. The visceral-to-gluteofemoral fat ratio was then calculated as the ratio of visceral to gluteofemoral fat volume for a given individual. Association analyses were adjusted for the same covariates described in the exome-wide discovery analysis, except for the exclusion of fine-mapped common alleles and the inclusion of height as additional covariate.

DXA was performed on ~5000 participants in UKB by General Electric Lunar iDXA instruments[69]. Scans were analyzed by the radiographer at image acquisition using General Electric enCORE software to generate all numerical indices of body composition (e.g. lean and fat mass). We derived the visceral-abdominal to gluteofemoral fat mass ratio and leg fat percentage from the DXA data. The protocol used for image acquisition is available at: https://biobank.ndph.ox.ac.uk/ukb/ukb/docs/DXA_explan_doc.pdf.

### Exome sequencing and genotyping data

The Regeneron Genetics Center (RGC) performed high coverage whole-exome sequencing in all cohorts. These procedures have been described in detail previously[22,63,81] and are briefly summarized here. To capture exome sequences, we used NimbleGen VCRome probes from Roche (for a fraction of GHS participants) or a modified version of the xGen design from Integrated DNA Technologies (IDT; for the remaining participants in GHS and all other cohorts). Next, we sequenced balanced pools using 75 base pair paired-end reads, using Illumina v4 HiSeq 2500 (for the initial part of the GHS cohort) or Illumina NovaSeq (for all other samples) instruments. We achieved more than 20x coverage over 85% of targeted bases in 96% of the VCRome-captured samples and 20x coverage over 90% of targeted bases in 99% of the IDT samples. We used Illumina software to demultiplex pooled samples following sequencing, used BWA-mem[82] to align reads to the GRCh38 human reference genome, and used GLnexus[83] to produce cohort-level genotype files. We used the snpEff[84] software and Ensembl v85 gene definitions to annotate variants. Annotations for protein-coding transcripts were prioritized using the most deleterious functional effect for each gene (ordered from most deleterious to least deleterious): frameshift, stop-gain, stop-loss, splice acceptor, splice donor, in-frame indel, missense, and other annotations. Predicted loss-of-function (pLOF) genetic variants included the following: (1) deletions or insertions resulting in a frameshift; (2) deletions, insertions, or single nucleotide variants resulting in the loss of a transcription start/stop site or introduction of a premature stop codon; and (3) acceptor or donor splice site variants. Missense variants were classified according to their predicted deleteriousness by way of several in silico algorithms. These were LRT[85], MutationTaster[86], SIFT[87], Polyphen2 HDIV[88] and Polyphen2 HVAR[88]. We then constructed seven gene-burden models for each gene, according to the functional annotation and alternative allele frequency (AAF) of each variant in that gene. This included: (1) pLOF variants only, AAF < 1%; (2) pLOF variants or missense variants predicted to be deleterious by all 5 in silico algorithms (as outlined above), AAF < 1%; (3) pLOF or missense variants predicted to be deleterious by all 5 in silico algorithms, AAF < 0.1%; (4) pLOF or missense variants predicted to be deleterious by at least 1 of 5 in silico algorithms, AAF < 1%; (5) pLOF or missense variants predicted to be deleterious by at least 1 of 5 in silico algorithms, AAF < 0.1%; (6) pLOF or

any missense variants (irrespective of predicted deleteriousness), AAF < 1%; 7) pLOF or any missense variants, AAF < 0.1%.

UKB generated genotyping array data as previously outlined[89]. We used the Illumina Human Omni Express Exome or Global Screening arrays[22] to perform common-variant genotyping in other cohorts. Variants were subsequently imputed separately according to genotyping platform, and using the TOPMed reference panel[90], via the TOPMed imputation server[91].

### Common-variant genome-wide association study and fine-mapping

We leveraged more than 9 million imputed common variants (minor allele frequency >1%) to conduct GWAS of BMI-adjusted WHR in UKB, MDCS, and MCPS. Association analyses were performed separately in each cohort and ancestry, using mixed-effects linear regression models implemented in REGENIE[92]. Ancestry-specific results were subsequently pooled across cohorts using fixed-effect, inverse-variance-weighted meta-analysis. Subsequently, we used the FINEMAP software to pinpoint the most likely causal variants for each genome-wide significant signal ($p < 5 \times 10^{-8}$ [22]). For this analysis, we defined loci as 1 MB windows centered on the variant with the smallest p-value at a locus. If association signals extended beyond this window, we expanded the window for 250 kb beyond variants with $p < 5 \times 10^{-5}$. Overlapping loci were merged into the same locus. Linkage disequilibrium was calculated for each locus using the same subjects included in the genome-wide association analysis, followed by fine-mapping (separately in each ancestry) implemented in FINEMAP[93]. At each locus, fine-mapping identifies sets of common variants (termed "credible sets") that have a high likelihood of including the causal variant at that locus. Each variant in a credible set is assigned a posterior inclusion probability (PIP), with a larger PIP representing a greater likelihood of a variant being the causal variant for that signal. We identified the 95% credible set (i.e., the smallest set of variants that captures 95% of the PIP) for each locus and assigned the variant with the highest PIP as the sentinel variant. Fine-mapping in the HLA region was approximated by identification of independent sentinel variants using linkage disequilibrium clumping, as implemented in Plink[94] (using the command "--clump --clump-r2 0.01 --clump-p1 5e-8 --clump-p2 5e-8").

### Exome-wide association analysis

We estimated the association between gene-burden models and phenotypes using linear regression (quantitative traits) or Firth-bias corrected logistic regression (binary outcomes), implemented in REGENIE[92]. Analyses were stratified by ancestry and adjusted for several covariates, including age, age$^2$, sex, age-by-sex and age$^2$-by-sex interaction terms, experimental batch-related covariates, the first ten common-variant-derived principal components (only four common-variant principal components were used in MCPS to account for specific level of admixture and relatedness in that study), and the first 20 rare-variant-derived principal components. We further adjusted discovery exome-wide analyses of BMI-adjusted WHR for common-variant signals identified by FINEMAP (identified as described above and listed in Supplementary Data 2), to ensure independence between rare and common-variant signals, as done previously[22]. We used fixed-effect inverse-variance-weighted meta-analysis to pool results across subsets and applied a Bonferroni-corrected statistical significance threshold of $p < 3.6 \times 10^{-7}$ in the gene-burden discovery analysis.

In a secondary analysis, we performed an exome-wide association analysis of BMI-adjusted WHR for individual rare nonsynonymous variants (minor allele frequency < 1% and minor allele count > 25) using the same analytical approach as for the gene-burden analysis and applying a statistical significance threshold of $p < 5 \times 10^{-8}$, as described before.[22]

## Tissue enrichment analysis

We calculated tissue enrichment for genes identified in the primary discovery analysis using gene expression values from the V8 data freeze from GTEx (https://www.gtexportal.org), as previously described [22].

## Identification of genes and variants associated with BMI-adjusted WHR

We sought to identify genes and variants for which the association with BMI-adjusted WHR had not been previously reported in large-scale rare coding variant association studies of this trait[17,18,29]. We extracted reported variants and genes from these studies meeting the following criteria: pLOF or missense variants (or gene burden of such variants), AAF < 1%, and p-value for association with BMI-adjusted WHR meeting conventional statistical significance thresholds ($p < 5 \times 10^{-8}$ for single rare coding variants, $p < 3.6 \times 10^{-7}$ for gene-burden analyses).

## Leave-one-variant-out backward-selection analysis

We used a leave-one-variant-out analysis to generate the list of individual variant sites that contribute to the observed association with lower BMI-adjusted WHR for rare coding variants in *INHBE*. In successive iterations, we identified the variant site whose removal maximally attenuated the gene-burden association signal (i.e., resulting in the largest p-value for association). In the following iteration, the identified variant was removed from the gene-burden. This was repeated until the gene-burden test based on the remaining list of variants had an association *p*-value > 0.05.

## Analysis of *INHBE* mRNA expression in the liver

Liver mRNA expression of *INHBE* was measured in 2,611 patients of the GHS bariatric cohort in whom RNA was sequenced on Illumina Nova-Seq instruments by 75 bp paired-end reads. The gene expression values for all samples were then normalized across samples using the trimmed mean of *m*-values approach (TMM) as implemented in edgeR[95,96]. To assess differential expression for *INHBE* expression among samples with various NAFLD Activity Scores (NAS) and among samples with different liver histopathology categories, we used DESeq2[97] with age, sex, race, and extraction site as covariates. We performed log fold change shrinkage between group comparisons using the 'apeglm' method[98] to achieve a more effective ranking across groups for *INHBE*, estimating a more precise log fold change.

## Expression of *INHBE* variants and immunoblotting of INHBE protein

*INHBE* wild type (WT) and c.299-1G>C expression constructs consisted of minigenes containing the full untranslated regions (UTRs), both exons, and the intron between exons 1 and 2, and were synthesized into the pcDNA3.1 vector. Commonly used hepatocyte cell lines such as HepG2 hepatoma cells express *INHBE* endogenously (Human Protein Atlas[32]). To ensure examination of only the *INHBE* splice variant and WT control, we expressed the *INHBE* WT and c.299-1G>C variant constructs in ExpiCHO-S cells, which do not express endogenous *INHBE*. Expression experiments were performed according to manufacturer's instructions (Thermofisher, A29133). Briefly, ExpiCHO-S cells were seeded at $3 \times 10^6$ cells/mL one day before transfection and transfected with 1 μg/mL *INHBE* plasmids on the day of transfection. ExpiCHO enhancer/feeder mixture was added 20 h after transfections. Cultures were harvested 3 days after transfections.

For immunoblotting of INHBE protein, cells were lysed in RIPA lysis buffer (Thermofisher, 89900) containing protease and phosphatase inhibitors (Thermofisher, 78441). Cell lysates, conditioned medium, and 100 ng of GST-tagged full length INHBE recombinant protein (Abnova, H00083729-P01) were run on SDS-PAGE under reducing conditions and transferred to PVDF membranes. Membranes were blocked in Superblock T20 TBS buffer (Thermofisher, 37536) then incubated in primary antibody against INHBE (Novus Biologicals, H00083729-B01P, 1:1000) overnight at 4 °C. Secondary antibody incubation was performed with HRP conjugated anti-mouse antibody (Cell Signaling, 7076, 1:10000) for 3 h at room temperature. Super-signal West Pico Plus Chemiluminescent Substrate (Thermofisher, 34579) was used for the development of chemiluminescent signal. Ponceau S (Sigma, P7170) was used to visualize total protein bands.

## Mendelian randomization analysis of fat distribution and liver traits

We examined the association between genetically predicted fat distribution or BMI and various liver and metabolic traits using Mendelian randomization (MR)[99]. We used the fixed-effect inverse-variance-weighted (IVW) method, implemented in the TwoSampleMR[100] and MendelianRandomization[101] R packages.

## Polygenic score analyses

We generated and evaluated polygenic scores that capture the common-variant-driven genetic predisposition to higher or lower BMI-adjusted WHR. We used genome-wide association analyses of imputed common variants in 461,548 European ancestry participants from UKB as the training dataset, and a non-overlapping sample of 24,958 unrelated European ancestry participants from the MDCS as a model selection and validation dataset. We generated polygenic scores using four different derivation approaches, after subsetting results to variants with a minor allele frequency ≥1%: (a) "clumping and thresholding"[94] using four different $r^2$ thresholds for linkage disequilibrium clumping (0.2, 0.4, 0.6, 0.8) at seven different p-value thresholds ($5 \times 10^{-02}$, $5 \times 10^{-03}$, $5 \times 10^{-04}$, $5 \times 10^{-05}$, $5 \times 10^{-06}$, $5 \times 10^{-07}$, $5 \times 10^{-08}$) for variant inclusion; (b) the LDpred algorithm, at ten different rho values (1, 0.1, 0.01, 0.001, 0.3, 0.03, 0.003, 0.00427, 0.00573, 0.00759); (c) conditional and joint analysis (COJO) approach, implemented in GCTA[102], at two p-value thresholds ($5 \times 10^{-07}$ and $5 \times 10^{-08}$) which uses a stepwise model selection approach for all variants that meet the selected *p*-value threshold; and (d) sBayesR[103] using its default parameters. Hence, a total of 41 different models. We selected the optimized polygenic score across the different approaches based on which method maximized the variance explained ($R^2$) for BMI-adjusted WHR in the model selection dataset. $R^2$ estimates were obtained using models that accounted for 10 common-variant genetic PCs, age, and sex. Using the model that yielded the optimized polygenic score out of the 41 mentioned above (COJO approach with a P-value threshold of $5 \times 10^{-07}$, using 500 variants), we generated the polygenic scores for BMI-adjusted WHR in the GHS and UKB cohorts. This polygenic score differs from the previously published polygenic scores for BMI-adjusted WHR used in the Mendelian randomization analyses described above. The polygenic scores used for Mendelian randomization have been validated as suitable instruments for the Mendelian randomization framework and have been built with the goal of facilitating etiologic inference (for instance by minimizing between-variant linkage disequilibrium). Conversely, the polygenic score generated and validated here aimed at maximizing variance explained and the ability to predict the BMI-adjusted WHR phenotype. To validate our approach, we performed a similar analysis using an independent GWAS training set of 142,762 people[14].

## Selection of high-impact genotypes

We compared the phenotypic impact of polygenic extreme and rare mutations with Mendelian-size effects. To define polygenic extreme, we used the top 1% or the bottom 1% of the polygenic score distribution, which has been shown to impart a phenotypic impact comparable to that of large-effect, rare variants[40,104]. We defined several further genotype groups. The first group was that of carriers of rare (AAF < 1%) pLOF or experimentally validated LOF variants in *PPARG*, the causal gene for FPLD type 3 and one of the genes discovered in our gene-

burden analysis, as a benchmark for a Mendelian-size effect in a WHR-increasing direction. Experimentally validated LOF variants were defined as missense variants predicted to be causal for FPLD type 3 based on a systematic functional characterization of all possible missense variants in *PPARG* and calibration with true FPLD type 3-causing mutations[105]. We used missense sites whose predicted probability of being causal for FPLD type 3 using this method was above 80%. Additional genotype groups included (1) individuals in the top quintile of the polygenic score distribution who were also heterozygous carriers of *ANKRD12* rare pLOF (since the burden of rare pLOF variants in *ANKRD12* had the largest WHR-increasing effect among genes discovered in our gene-burden analysis); (2) *PLIN4* pLOF homozygotes, since *PLIN4* was the only gene in our gene-burden discovery analysis for which the rare pLOF variant burden was associated with a large WHR-increasing effect (>0.1 SD) and for which multiple pLOF homozygotes were identified (i.e. complete human "knock-outs"); (3) *INHBE* pLOF carriers, as a benchmark for a rare-variant-driven effect in a WHR-decreasing direction; (4) individuals who were *INHBE* pLOF carriers and who were also in the bottom quintile of the polygenic score distribution.

### Reporting summary

Further information on research design is available in the Nature Research Reporting Summary linked to this article.

## Data availability

The data supporting the findings of this manuscript are reported in the main text, in the figures, in the supplementary materials, and are tabulated in Table 1, Table 2 and Supplementary Data 1 to 28. UKB individual-level genotypic and phenotypic data may be accessed by approved investigators via the UK Biobank study (www.ukbiobank.ac.uk/). Additional information about registration for access to the data are available at www.ukbiobank.ac.uk/register-apply/. Data access for approved applications requires a data transfer agreement between the researcher's institution and UK Biobank, the terms of which are available on the UK Biobank website (www.ukbiobank.ac.uk/media/ezrderzw/applicant-mta.pdf). MCPS data may be available to qualified non-commercial researchers to reproduce results reported in this manuscript by emailing mcps-access@ndph.ox.ac.uk. The data access policy can be downloaded from https://www.ctsu.ox.ac.uk/research/prospective-blood-based-study-of-150-000-individuals-in-mexico.

MDCS data may be available to qualified academic non-commercial researchers to reproduce results reported in this manuscript through the portal at https://www.malmo-kohorter.lu.se/malmo-cohorts, following the principles outlined in this policy https://www.malmo-kohorter.lu.se/sites/malmo-kohorter.lu.se/files/mdcs_mpp_mos_request_form_vermar20.doc. eQTL summary statistics may be downloaded from the GTEx portal (https://gtexportal.org/). The GRCh38 reference assembly may be accessed from the Genome Reference Consortium (https://www.ncbi.nlm.nih.gov/grc).

## Code availability

The REGENIE association analysis package was used to perform genetic associations (available at https://doi.org/10.5281/zenodo.6789127). Reads were aligned to the GRCh38 reference genome using BWA-mem[82] and GLnexus[83] was used to produce cohort-level genotype files. Variants were annotated with snpEff[84]. Missense variants were annotated using LRT[85], MutationTaster[86], SIFT[87], Polyphen2 HDIV[88] and Polyphen2 HVAR[88]. Fine-mapping of GWAS data was performed using FINEMAP v1.4[93] and variants in the HLA region were clumped using Plink[94]. Polygenic scores were derived using GCTA v1.93[102], LDpred v1.0.11[106], and sBayesR v2.02[103]. Mendelian randomization analyses were performed using TwoSampleMR v0.5.6[100] and MendelianRandomization v0.5.1[101]. Liver gene expression data were analyzed using edgeR v3.32.1, DESeq2[97], and apeglm v1.12.0[98].

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

## Acknowledgements

This research was funded by Regeneron Pharmaceuticals. The Malmö Diet and Cancer study was funded by grants from the Swedish Medical Research Council, the Swedish Cancer Foundation, the Albert Påhlsson and Gunnar Nilsson Foundations, AFA insurance, and the Malmö city council. Mexico City Prospective Study is funded by a core grant from the UK Medical Research Council (MC_UU_00017/2) to the MRC Population Health Research Unit at the University of Oxford, and has previously received funding from the Mexican Health Ministry, the Mexican National Council of Science and Technology, the Wellcome Trust, the British Heart Foundation, Cancer Research UK, and the Nuffield Department of Population Health at the University of Oxford. O.M., A.G., M.O.M. received funding from the European Research Council (ERC-AdG-2019-885003).

## Author contributions

L.A.L., N.V., A.B., designed the study; P.A., O.A.S., J.Bovijn, N.V., L.A.L. wrote the first draft of the manuscript. P.A., O.A.S., J.Bovijn, K.L., J.B.N., M.K., S.A., T.D., M.E.H., G.H., N.L., I.R.D., J.Z.L., S.H., B.G., M.G., L.P., P.P., J.R.W., G.H., G.S.A., M.J., M.G.L., C.D.S., D.J.C., A.G., M.O.M., J.Berumen, P.K.M., J.A.D., J.M.T., J.R.E., R.C., D.J.R., B.Z., A.J.M., S.B., J.D.O., J.G.R., A.R.S., M.C., G.R.A., M.A.R.F., M.W.S., V.G., J.A., C.H., A.N.E., V.I., K.K.,

G.D.G., T.M., G.D.Y., O.M., J.M., R.T.C., A.E.L., A.B., N.V., L.A.L participated in data collection, analysis and/or interpretation, reviewed the manuscript for important intellectual content, and agreed to the decision to submit the manuscript for publication.

## Competing interests

Regeneron authors receive salary from and own options and/or stock of the company. G.D.Y is the Chief Scientific Officer and member of the Board of Directors at Regeneron Pharmaceuticals; A.J.M is an Executive Officer of Regeneron Pharmaceuticals. L.A.L., P.A., O.S., M.A.R.F., and A.B. are inventors on provisional patent applications (63/233,258 and 63/274,595), U.S. non-provisional applications (17/549,692, and 17/711,137), and PCT international application (PCT/US21/63150) submitted by RGC relating to *INHBE* genetics. N.V., O.S., P.A., A.L., A.B., and L.A.L. are inventors on U.S. non-provisional applications (17/740,382), and PCT international application (PCT/US22/28415) submitted by RGC relating to *PDE3B* genetics. Other co-authors did not declare competing interests.

## Additional information

[1]Regeneron Genetics Center, Regeneron Pharmaceuticals Inc, Tarrytown, NY, USA. [2]Regeneron Pharmaceuticals Inc, Tarrytown, NY, USA. [3]Department of Molecular and Functional Genomics, Geisinger Health System, Danville, PA, USA. [4]Geisinger Obesity Institute, Geisinger Health System, Danville, PA, USA. [5]Department of Clinical Sciences Malmö, Lund University, Malmö, Sweden. [6]Department of Medicine, University of Verona, Verona, Italy. [7]Unidad de Medicina Experimental de la Facultad de Medicina de la Universidad Nacional Autónoma de México, Mexico City, Mexico. [8]Instituto Tecnológico y de Estudios Superiores de Monterrey, Monterrey, Mexico. [9]MRC Population Health Research Unit, Nuffield Department of Population Health, University of Oxford, Oxford, UK. [10]Clinical Trial Service Unit & Epidemiological Studies Unit Nuffield Department of Population Health, University of Oxford, Oxford, UK. [11]Department of Genetics, Perelman School of Medicine, University of Pennsylvania, Philadelphia, PA, USA. [12]Department of Emergency and Internal Medicine, Skåne University Hospital, Malmö, Sweden. [13]These authors contributed equally: Parsa Akbari, Olukayode A. Sosina, Jonas Bovijn, Roberto Tapia-Conyer, Adam E. Locke, Aris Baras, Niek Verweij, Luca A. Lotta. ✉e-mail: aris.baras@regeneron.com; luca.lotta@regeneron.com

## Regeneron Genetics Center

**Parsa Akbari**[1,13]✉, **Olukayode A. Sosina**[1,13], **Jonas Bovijn**[1,13], **Jonas B. Nielsen**[1], **Minhee Kim**[1], **Senem Aykul**[1], **Tanima De**[1], **Mary E. Haas**[1], **George Hindy**[1], **Nan Lin**[1], **Benjamin Geraghty**[1], **Marcus Jones**[1], **Michelle G. LeBlanc**[1], **Suganthi Balasubramanian**[1], **John D. Overton**[1], **Jeffrey G. Reid**[1], **Alan R. Shuldiner**[1], **Michael Cantor**[1], **Goncalo R. Abecasis**[1], **Manuel A. R. Ferreira**[1], **Aris N. Economides**[1,2], **Katia Karalis**[1], **Giusy Della Gatta**[1], **Jonathan Marchini**[1], **Adam E. Locke**[1,13], **Aris Baras**[1,13], **Niek Verweij**[1,13] & **Luca A. Lotta**[1,13]✉

## DiscovEHR Collaboration

**Ian R. Dinsmore**[3], **Jonathan Z. Luo**[3], **Christopher D. Still**[4], **David J. Carey**[4] & **Tooraj Mirshahi**[4]

Full list of members and their affiliations appears in the Supplementary Information.

