## [Peer Review File · Nature Communications]

Multiancestry exome sequencing reveals INHBE mutations associated with favorable fat distribution and protection from diabetesREVIEWER COMMENTS

Reviewer #1 (Remarks to the Author):

In this study the authors used an interesting approach by combining large-scale exome sequencing with anthropometric and metabolic phenotyping in over 600,000 people, to investigate genetic determinants of the ability to favorably store fat in appropriate peripheral adipose depots. Then they studied whether the variability in the identified genes also associates with a re-distribution of the lipids to the liver, with severity of the liver phenotype and with type 2 diabetes. The authors found that polygenic common variation and rare coding variants, including a novel association for rare predicted loss-of-function variants in the liver-specific *INHBE*, are associated with favorable fat distribution, lower atherogenic lipids, lower markers of liver inflammation and damage, and protection from type 2 diabetes. Interestingly, in this respect, they observed statistically significant enrichment for genes that are highly expressed in subcutaneous adipose tissue, and, to a lesser degree, visceral adipose tissue. They also found that common variants or rare coding alleles that are linked with an impaired capacity to expand peripheral adipose tissue to be associated with higher ectopic fat deposition in the liver and hepatic inflammation. In addition, the authors found that individuals at the extreme of the newly identified polygenic predisposition to higher WHRadjBMI have a similar fat distribution phenotype as carriers of *PPARG* mutations, but with an over 100-fold more frequent genotype than these rare mutations.

This is a very well performed large translational study providing very interesting novel findings in an important field of research.

Comments:

1. Table S8: the authors show the relationships of the 16 genes specifically with waist and with hip circumferences after adjustment of these measures for BMI. The authors should also test these relationships separately in women and men.
2. In the title the authors should also provide information about important cardiometabolic clinical endpoints of their study, e.g. 'Multi-ancestry exome sequencing reveals *INHBE* protein-truncating variants associated with favorable fat distribution, liver-adipose interplays in energy storage and protection from type 2 diabetes'.
3. What may upregulate *INHBE* expression in NASH?

Reviewer #2 (Remarks to the Author):

Akbari and colleagues have conducted an analysis of waist-to-hip ratio in whole exome sequences from up to six hundred thousand individuals. They identify sixteen genes for which rare coding variants associate with WHRadjBMI at exome wide significance. They identify rare protective LOF variants in *INHBE*, a circulating ligand, that associated with improved body fat distribution and nominally lower risk of type 2 diabetes and liver injury.

I have only minor comments on this exceptional study.

1. I would note that, although six of the seven known FPLD associate with fat distribution, the direction of the associations is not consistent with the reported Mendelian phenotypes. pLOF in *PLIN1*, in particular, associate with reduced WHRadjBMI in the current analysis but were previously reported to be causative for partial lipodystrophy (Gandotra NEJM). These findings suggest that *PLIN1* knockdown may be a therapeutic approach for lipodystrophy and that the previously identified frameshift variants are likely gain-of-function.
2. Any association of the *INHBE* variants with fracture risk or bone mineral density? On global biobank engine the *INHBE* splice variant associates with increased risk of fracture of several sites (https://biobankengine.stanford.edu/RIVAS_HG19/variant/12-57849876-G-C) which is also observed

clinically with PPARG agonists and with the damaging ALK7 variants.

3. INHBE LOF variants appear to strongly associate with increased birth weight -

<https://azphewas.com/geneView/7e2a7fab-97f0-45f7-9297-f976f7e667c8/INHBE/glr/continuous>.

4. If possible, I would show the effect of the INHBE variants on gluteofemoral, ASAT and VAT as well as impedance based measurements. Given than the INHBE LOF variant associates with elevated BMI, it's likely that they associate with increased peripheral body fat but still protect against type 2 diabetes, liver fat etc. I would similarly show the effect of the polygenic score on total peripheral fat. The fact that the protective variants at PDE3B, ALK7 and now INHBE increase peripheral fat and body fat percentage (and not just shift fat from visceral depots to periphery) is not widely appreciated.

Reviewer #1

Reviewer 1 Opening Statement (R1OS): “In this study the authors used an interesting approach by combining large-scale exome sequencing with anthropometric and metabolic phenotyping in over 600,000 people, to investigate genetic determinants of the ability to favorably store fat in appropriate peripheral adipose depots. Then they studied whether the variability in the identified genes also associates with a re-distribution of the lipids to the liver, with severity of the liver phenotype and with type 2 diabetes. The authors found that polygenic common variation and rare coding variants, including a novel association for rare predicted loss-of-function variants in the liver-specific *INHBE*, are associated with favorable fat distribution, lower atherogenic lipids, lower markers of liver inflammation and damage, and protection from type 2 diabetes. Interestingly, in this respect, they observed statistically significant enrichment for genes that are highly expressed in subcutaneous adipose tissue, and, to a lesser degree, visceral adipose tissue. They also found that common variants or rare coding alleles that are linked with an impaired capacity to expand peripheral adipose tissue to be associated with higher ectopic fat deposition in the liver and hepatic inflammation. In addition, the authors found that individuals at the extreme of the newly identified polygenic predisposition to higher WHRadjBMI have a similar fat distribution phenotype as carriers of *PPARG* mutations, but with an over 100-fold more frequent genotype than these rare mutations.

This is a very well performed large translational study providing very interesting novel findings in an important field of research.”

R1OS - Author’s reply: We thank the Reviewer for their thorough evaluation of our work and the very helpful comments. We have revised the manuscript accordingly and find that it has greatly improved as a result. A point-by-point reply to the Reviewer’s comments can be found below.

Reviewer 1 Comment 1 (R1C1): “Table S8: the authors show the relationships of the 16 genes specifically with waist and with hip circumferences after adjustment of these measures for BMI. The authors should also test these relationships separately in women and men.”

R1C1 - Author’s reply and manuscript changes: We thank the Reviewer for their helpful comment and have added the sex-stratified estimates to **Table S8** in the revised manuscript.

R1C2: “In the title the authors should also provide information about important cardiometabolic clinical endpoints of their study, e.g. ‘Multi-ancestry exome sequencing reveals *INHBE* protein-truncating variants associated with favorable fat distribution, liver-adipose interplays in energy storage and protection from type 2 diabetes’.”

R1C2 - Author’s reply and manuscript changes: We thank the Reviewer for this excellent suggestion. We have now changed the title to “*Multi-ancestry exome sequencing reveals INHBE protein-truncating variants associated with favorable fat distribution and protection from type 2 diabetes*”, which highlights the protective association with type 2 diabetes.

R1C3: “What may upregulate INHBE expression in NASH?”

R1C3 - Author’s reply and manuscript changes: This is a very interesting question. In the limited literature on the role of *INHBE* in humans, there is a suggestion that hepatic *INHBE* expression is upregulated in obese people with insulin resistance (*PLoS One*. 2018;13:e0194798). This suggests that insulin plays a role in *INHBE* regulation, and it is possible that the higher *INHBE* expression in steatosis, NASH and fibrosis may reflect the underlying hyperinsulinemia of those conditions. In turn, we hypothesize that the upregulation of *INHBE* may contribute to the metabolic dysfunction of those states by further inhibiting adipose storage. Future studies are necessary to shed further light on the broader influences of metabolic pathways on *INHBE* expression. We have added mention of these points in the **Discussion** section of the revised manuscript, Page 15, Line 23: “*Higher levels of INHBE expression have been observed in insulin resistance (52), an early pathophysiologic process in metabolic disease, and, in our study, in hepatic steatosis and inflammation, which may partly reflect the insulin resistance associated with those conditions. We hypothesize that the upregulation of INHBE in those settings may drive a maladaptive response to excess calories.*”.

Reviewer #2

R2OS: “Akbari and colleagues have conducted an analysis of waist-to-hip ratio in whole exome sequences from up to six hundred thousand individuals. They identify sixteen genes for which rare coding variants associate with WHRadjBMI at exome wide significance. They identify rare protective LOF variants in *INHBE*, a circulating ligand, that associated with improved body fat distribution and nominally lower risk of type 2 diabetes and liver injury.

I have only minor comments on this exceptional study.”

R2OS - Author’s reply: We thank the Reviewer for their evaluation of our work and for their thoughtful comments. We have revised the manuscript accordingly. A point-by-point reply to the Reviewer’s comments can be found below.

R2C1: “I would note that, although six of the seven known FPLD associate with fat distribution, the direction of the associations is not consistent with the reported Mendelian phenotypes. pLOF in *PLIN1*, in particular, associate with reduced WHRadjBMI in the current analysis but were previously reported to be causative for partial lipodystrophy (Gandotra NEJM). These findings suggest that *PLIN1* knockdown may be a therapeutic approach for lipodystrophy and that the previously identified frameshift variants are likely gain-of-function.”

R2C1 - Author’s reply and manuscript changes: We thank the Reviewer for this comment and agree that this is an interesting observation. We now discuss this in the **Results** section of the revised manuscript (Page 7, Line 2) and also explore the difference with previously reported frameshift variants in a dedicated supplementary results section (**Results S1**).

R2C2: “Any association of the *INHBE* variants with fracture risk or bone mineral density? On global biobank engine the *INHBE* splice variant associates with increased risk of fracture of several sites (https://biobankengine.stanford.edu/RIVAS_HG19/variant/12-57849876-G-C) which is also observed clinically with *PPARG* agonists and with the damaging *ALK7* variants.”

R2C2 - Author’s reply and manuscript changes: We agree that this is an important and therapeutically-relevant question. We have examined the association of rare coding variants in *INHBE* and *PPARG* with estimated bone mineral density and fracture risk. We found no evidence of association between rare pLOF variants in *INHBE* with either estimated bone mineral density or fracture risk. In contrast, and consistent with the clinical evidence for *PPARG* agonists, we found that rare predicted loss-of-function and deleterious missense variants in *PPARG* are associated with fracture risk. These data are now described in the **Results** section of the revised manuscript (Page 10, Line 19), in a dedicated supplementary results section (**Results S3**) and in **Table S28**.

R2C3: “*INHBE* LOF variants appear to strongly associate with increased birth weight - <https://azphewas.com/geneView/7e2a7fab-97f0-45f7-9297-f976f7e667c8/INHBE/glr/continuous>.”

R2C3 - Author’s reply and manuscript changes: This is an interesting observation. We now include the association estimates with birth weight in the **Results** section (Page 9, Line 7) and in **Table S13**.

R2C4: “If possible, I would show the effect of the *INHBE* variants on gluteofemoral, ASAT and VAT as well as impedance based measurements. Given than the *INHBE* LOF variant associates with elevated BMI, it’s likely that they associate with increased peripheral body fat but still protect against type 2 diabetes, liver fat etc. I would similarly show the effect of the polygenic score on total peripheral fat. The fact that the protective variants at *PDE3B*, *ALK7* and now *INHBE* increase peripheral fat and body fat percentage (and not just shift fat from visceral depots to periphery) is not widely appreciated.”

R2C4 - Author’s reply and manuscript changes: We thank the Reviewer for these excellent suggestions. In the revised manuscript, we have included results for the association of *INHBE* pLOF variants with MRI-based and bioelectrical impedance measures (Page 9, Line 4 of the **Results** section, **Table S12** and **Figure S6**). We have noted the impact on hip circumference as a proxy measure for peripheral fat for rare variants in *INHBE*, *PDE3B* and *ACVR1C* (*ALK7*) in the **Results** section (Page 10, Line 9) and **Table S8**. We also now show the impact of polygenic scores on leg fat mass and percentage as key measures of peripheral fat depots in **Figure S14**.